# LycheeDecode: Accelerating Long-Context LLM Inference via Hybrid-Head Sparse Decoding

**Gang Lin, Dongfang Li, Zhuoen Chen, Yukun Shi, Xuhui Chen, Baotian Hu & Min Zhang**
Harbin Institute of Technology, Shenzhen
`glin3104@gmail.com`
`{lidongfang, hubaotian, zhangmin2021}@hit.edu.cn`

## Abstract

The proliferation of long-context large language models (LLMs) exposes a key bottleneck: the rapidly expanding key-value cache during decoding, which imposes heavy memory and latency costs. While recent approaches attempt to alleviate this by sharing a single set of crucial tokens across layers, such coarse-grained sharing undermines model performance by neglecting the functional diversity of attention heads. To address this, we propose LycheeDecode, an efficient decoding method centered on a fine-grained hybrid-head attention mechanism that employs a hardware-efficient top-$k$ selection strategy. Specifically, the novel HardKuma-based mechanism partitions attention heads into a small subset of retrieval heads that dynamically identify crucial tokens and a majority of sparse heads that reuse them for efficient computation. Through extensive experiments on leading models like Llama3 and Qwen3 across diverse benchmarks for long-context understanding (e.g., LongBench, RULER) and complex reasoning (e.g., AIME24, OlympiadBench), we demonstrate that LycheeDecode achieves generative quality comparable to, and at times surpassing even the full-attention baseline. Crucially, this is accomplished with up to a $2.7\times$ speedup at a 128K context length. By preserving the functional diversity of attention heads, our fine-grained strategy overcomes the performance bottlenecks of existing methods, providing a powerful and validated pathway to both efficient and high-quality long-context LLM inference. Code is available at `https://github.com/HITsz-TMG/TMGNLP/tree/main/LycheeDecode`.

## 1 Introduction

Transformer-based Large Language Models (LLMs) now possess remarkable long-context capabilities. Leading models like GLM-4 (GLM et al., 2024), Qwen2.5-1M (Yang et al., 2025a) and Gemini-2.5 (Comanici et al., 2025) support up to 1 million tokens, enabling superior performance in various long-text tasks such as summarization (Huang et al., 2021), question answering (Wei et al., 2022), multi-turn dialogue (Li et al., 2025), and complex reasoning (Wang et al., 2024).

However, long-context processing is challenging. Due to the autoregressive nature of the Transformer, for each new token generated, the model must perform attention calculations with the full key-value (KV) cache of previous tokens, leading to frequent memory access and increased I/O overhead. As the sequence grows, the KV cache expands linearly, leading to a surge in memory usage and a significant increase in computational latency. This severely constrains the deployment and scalability of long-context language models in practical applications. To address this challenge, recent work has proposed sparse attention methods, which reduce computational overhead by computing attention on only a small subset of critical tokens, exploiting the inherent sparsity of the attention mechanism. Typically, these methods are categorized into two types: *eviction-based methods* (Xiao et al., 2024; Li et al., 2024; Zhang et al., 2023), which compress the KV cache by permanently discarding tokens, and *selection-based methods* (Gao et al., 2025; Yang et al., 2025b; Wu et al., 2025), which preserve the full KV cache while dynamically selecting a subset of tokens for computation at each inference step. A key observation is that recent work has identified a high

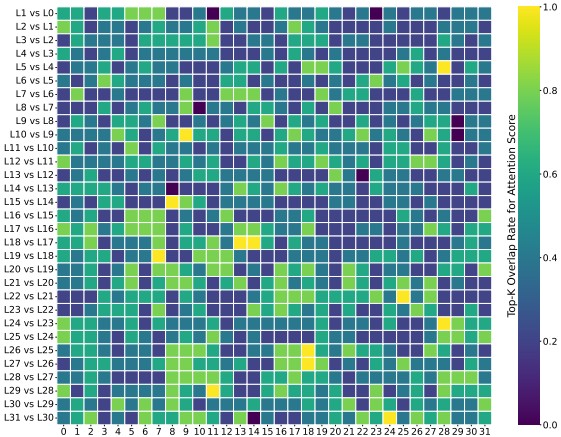

Figure 1: Overlap rate of top-$k$ ($k = 5$) attention scores between corresponding heads in adjacent layers. The heatmap illustrates the functional diversity among attention heads. We input prompt *Please directly output the final answer based on the given question. Question: In a world containing only squares, circles, and triangles, one shape is defined by having no angles and being perfectly symmetrical from every point on its perimeter. What is the single name of the only shape that fits this description? Answer:*, and Llama-3 outputs *circle*. More cases can be found in Appendix E.3.

degree of similarity in critical tokens across consecutive layers (Yang et al., 2025b; Hao et al., 2025). Consequently, they adopt a layer-level sharing strategy, where the same set of selected critical tokens is shared across all heads in subsequent layers. This hierarchical strategy forces all attention heads in the same layer to perform the same function. However, attention heads on the same layer do not exhibit highly similar patterns. As shown in Figure 1, the top-$k$ overlap rate of different heads in adjacent layers can vary significantly (e.g., the overlap rate of the 14th head of the last two layers is 0%, while the 24th head is 100%). **This suggests that a uniform, layer-wise sharing strategy may be overly simplistic, and a more fine-grained, head-based strategy is necessary.**

Inspired by this, we introduce **LycheeDecode**, a simple and effective hybrid-head sparse decoding method that refines this sharing strategy to a more granular level. Specifically, we classify attention heads into a few "retrieval heads" and a majority of "sparse heads". The retrieval heads are responsible for performing full attention computation over the entire context to accurately identify the most important tokens. This selected tokens are then shared with the sparse heads in subsequent layers for efficient sparse attention computation. In this way, LycheeDecode can capture more diverse and relevant attention patterns with minimal precision loss. On the other hand, identifying the types of attention heads typically involves optimizing a set of discrete binary variables. Previous work (Xiao et al., 2025) circumvents the challenge of discrete optimization by having each head learn a continuous variable. Although this variable is amenable to gradient-based methods during training, it must be rounded to a binary value for inference, which introduces a significant train-inference discrepancy that can degrade performance. To bridge this gap, we further introduce the Hard Kumaraswamy distribution (Kumaraswamy, 1980; Bastings et al., 2019). The HardKuma distribution is specifically designed to produce values that are naturally concentrated at 0 and 1, while remaining differentiable. By optimizing the distributional parameters of HardKuma during training, our model learns a near-binary selection mechanism directly, thus mitigating the train-inference discrepancy and leading to a more stable and robust head specialization. Evaluation with Llama3 and Qwen3 models on the long-context understanding (e.g., LongBench (Bai et al., 2024), RULER (Hsieh et al., 2024)) and complex reasoning (e.g., AIME24, OlympiadBench) tasks demonstrate that LycheeDecode can achieve the best performance among other methods with the same sparsity. It can also achieve $2.7\times$ the end-to-end decoding speedup compared to FlashAttention-2 implementation under 128k context length. Our contributions are summarized as follows:

- We propose LycheeDecode, a novel hybrid head sparse decoding method that delegates token selection to a small number of "retrieval heads", allowing for a more fine-grained and effective sparse attention mechanism.

- We introduce the Hard Kumaraswamy distribution to address the discrete optimization problem in end-to-end head type identification, reducing the train-inference gap and improving model robustness and performance.
- We implement the hybrid head block-sparse decoding kernel using TileLang (Wang et al., 2025), achieving up to $2.7\times$ end-to-end decoding speedup.

## 2  RELATED WORK

**Sparse attention methods**   These methods reduce computational and memory overhead during inference, falling into two main types: eviction-based and selection-based. Eviction-based sparse attention aims to lower KV cache memory usage by removing tokens considered less relevant (Xiao et al., 2024; Zhang et al., 2023; Li et al., 2024). In contrast, selection-based sparse attention preserves the full KV cache and selects the most important tokens for the attention mechanism to process (Gao et al., 2024; 2025; Bastings et al., 2019; Liu et al., 2024). Recent works explored trainable mechanisms to further refine token selection. Methods such as Native Sparse Attention (Yuan et al., 2025) and MiniCPM (Team et al., 2025) demonstrate that extensive post-training with sparse constraints can yield efficient decoding while maintaining high performance. These methods effectively balance performance with efficiency, mitigating the risk of information loss.

**Attention head functional specialization**   A key insight in long-context inference is the functional specialization of attention heads, with a small subset of "retrieval heads" being crucial for recalling information (Wu et al., 2025). Building on this, RazorAttention (Tang et al., 2025) introduced a training-free compression technique that exclusively maintains a full KV cache for these crucial retrieval heads while discarding remote tokens in other heads. DuoAttention (Xiao et al., 2025) and PruLong (Bhaskar et al., 2025) categorize heads as either "retrieval" or "streaming" by learning a continuous gating variable. However, these methods determine the role of each head in isolation, lacking a mechanism for direct collaboration. Unlike previous works, in our framework, retrieval heads not only perform full attention but also dynamically identify and propagate a curated subset of critical tokens for reuse by the majority of "sparse heads". **This creates a fine-grained, cooperative mechanism. It differs from previous methods by enabling more direct and efficient sharing of contextual information between functionally distinct heads.**

**Cross-layer attention similarity**   Recent studies have identified a high degree of similarity in important tokens and attention patterns across consecutive Transformer layers. This insight has inspired layer-level sharing strategies to improve inference efficiency. Approaches such as TidalDecode (Yang et al., 2025b) and OmniKV (Hao et al., 2025) designate a few selector layers to identify critical tokens, which are then reused by subsequent layers for efficient sparse computation. Other methods, like LiSA (Mu et al., 2024) and PoD (Ma et al., 2024), leverage this redundancy by directly sharing attention weights or key states across layers to reduce redundant calculations. However, their layer-level nature can overlook the functional diversity of individual attention heads. In contrast, our proposed LycheeDecode framework introduces a more fine-grained, head-level strategy, which preserves the functional diversity of attention heads, allowing for a more precise and adaptive mechanism by enabling more efficient sharing of contextual information.

## 3  METHODOLOGY

This section introduces LycheeDecode, a head-level sparse decoding framework that leverages the functional specialization of Transformer attention heads, as illustrated in Figure 2. LycheeDecode assigns heterogeneous roles to heads: **Retrieval Heads** that actively refresh critical tokens, and **Sparse Heads** that efficiently reuse them. By propagating token selections across layers, LycheeDecode improves efficiency while maintaining model performance.

### 3.1  HEAD-LEVEL SPARSE DECODING

**Retrieval Heads for Critical Token Identification.**   Certain attention heads are well-suited for capturing long-range dependencies such as co-reference resolution or distant contextual links. We

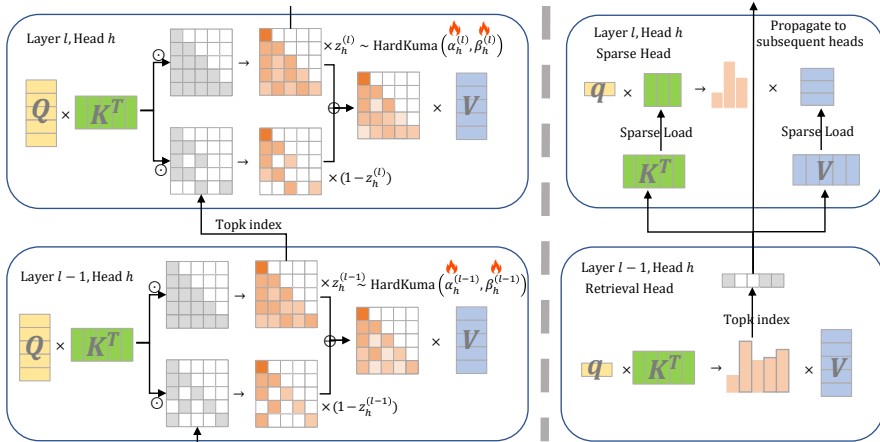

Figure 2: Overall framework. **Left**: During the training phase, each head calculates full attention and sparse attention, weighted by HardKuma sampling values. **Right**: During inference, the retrieval head calculates the critical tokens set for efficient calculation by the subsequent sparse heads.

designate these as Retrieval Heads ($h \in \mathcal{H}_R$). A Retrieval Head performs standard dense attention over the full sequence:

$$A_h^{(l)} = \text{softmax}\left( \frac{q_h^{(l)}(K_h^{(l)})^T}{\sqrt{d_k}} \right). \tag{1}$$

From the resulting attention map $A_h^{(l)}$, it selects the indices of the top-$k$ attended tokens:

$$\mathcal{S}_h^{(l+1)} = \text{argsTopK}(A_h^{(l)}, k), \tag{2}$$

where argsTopK returns the $k$ tokens with the highest attention weights. The updated set $\mathcal{S}_h^{(l+1)}$ is propagated to the head of the same index in the next layer, where it is then used by the subsequent attention heads for sparse attention computation. To initialize the critical token set $\mathcal{S}_h^{(0)}$, all heads in the first layer are designated as Retrieval Heads.

**Sparse Heads for Efficient Computation.** The other heads perform sparse attention computation on the critical token set, which we designate as Sparse Heads ($h \in \mathcal{H}_S$). A Sparse Head reuses the token set $\mathcal{S}_h^{(l)}$ inherited from the previous layer and restricts attention computation accordingly:

$$O_h^{(l)} = \text{softmax}\left( \frac{q_h^{(l)}(K_h^{(l)}[\mathcal{S}_h^{(l)}])^T}{\sqrt{d_k}} \right) V_h^{(l)}[\mathcal{S}_h^{(l)}], \tag{3}$$

where $K_h^{(l)}[\mathcal{S}_h^{(l)}]$ and $V_h^{(l)}[\mathcal{S}_h^{(l)}]$ denote the key and value matrices at head $h$ restricted to the subset of tokens indexed by $\mathcal{S}_h^{(l)}$. Since no new tokens are selected, the set is propagated unchanged, i.e., $\mathcal{S}_h^{(l+1)} = \mathcal{S}_h^{(l)}$. This mechanism reduces both the amount of computation and the KV-cache loading cost, which constitutes the dominant efficiency gain during autoregressive decoding.

**Retrieval–Sparse Synergy.** The interaction between Retrieval and Sparse Heads forms a decoding pipeline that is both adaptive and efficient. Retrieval Heads periodically refresh the salient token set, ensuring responsiveness to new context, while Sparse Heads exploit these curated subsets for efficient computation across layers. This division of labor allows LycheeDecode to trade off adaptivity and efficiency in a principled manner. The complete procedure is summarized in Appendix B.

### 3.2 HEAD SPECIALIZATION VIA HARDKUMA

The core challenge here lies in effectively classifying each attention head as either a Retrieval ($\mathcal{H}_R$) or a Sparse ($\mathcal{H}_S$) head. This assignment is fundamentally a discrete optimization problem over a set

of binary variables. Prior work, such as DuoAttention (Xiao et al., 2025), addresses this by learning a continuous variable for each head. Although this continuous variable is easily optimized, it must be rounded to a binary value for inference, which introduces the train-inference discrepancy.

To bridge this gap, our approach leverages the Hard Kumaraswamy (HardKuma) distribution (Kumaraswamy, 1980; Bastings et al., 2019), a differentiable proxy for binary variables. The HardKuma distribution is specifically designed to produce values that are naturally concentrated at 0 and 1, yet remains reparameterizable. By optimizing the distributional parameters of HardKuma during training, our model learns a near-binary selection mechanism directly, thus mitigating the train-inference discrepancy and leading to a more stable and robust head specialization.

**The HardKuma Distribution.** The HardKuma distribution provides a reparameterizable way to model near-binary choices. A sample $z \in [0, 1]$ is generated through a three-step process:

1. **Sample:** First, a sample $u$ is drawn from a uniform distribution, $u \sim \mathcal{U}(0, 1)$. Then using the inverse CDF of the Kumaraswamy distribution, $u$ is transformed into a sample $s = (1 - u^{1/\beta})^{1/\alpha}$, where $s \sim \text{Kuma}(\alpha, \beta)$.

2. **Stretch:** This sample $s \in (0, 1)$ is then linearly stretched to a wider interval $(p, q)$ where $p < 0$ and $q > 1$: $s' = s \cdot (q - p) + p$.

3. **Rectify:** Finally, $s'$ is passed through a hard-sigmoid function (i.e., clipping) to produce the final sample: $z = \min(1, \max(0, s'))$.

This process causes probability mass from the intervals $(p, 0]$ and $[1, q)$ to collapse at exactly 0 and 1, respectively, making the output near-binary while the entire transformation from $u$ remains differentiable almost everywhere.

**Identifying Attention Head Types.** To facilitate the learning of head roles, we introduce a differentiable training framework. Formally, for each head $h$ in layer $l$ (for $l > 0$), we associate a latent random variable $z_h^{(l)}$ sampled from a HardKuma distribution, governed by learnable $\alpha_h^{(l)}$ and $\beta_h^{(l)}$:

$$z_h^{(l)} \sim \text{HardKuma}(\alpha_h^{(l)}, \beta_h^{(l)}). \tag{4}$$

During training, each head computes attention maps for both potential roles to create a fully differentiable learning path. It generates a sparse attention map $A_{S,h}^{(l)}$ using an inherited token set $\mathcal{S}_h^{(l)}$, as well as a full attention map $A_{R,h}^{(l)}$. The full attention map is also used to select the token set $\mathcal{S}_h^{(l+1)}$ for the next layer (Equation 2). These two attention maps are linearly combined to form a single hybrid attention map $\tilde{A}_h^{(l)}$ using the stochastic sample $z_h^{(l)}$ as a weight:

$$\tilde{A}_h^{(l)} = z_h^{(l)} \cdot A_{R,h}^{(l)} + (1 - z_h^{(l)}) \cdot A_{S,h}^{(l)}. \tag{5}$$

It creates a fully differentiable path, allowing gradients from the final loss to flow back and update the distributional parameters $\alpha_h^{(l)}$ and $\beta_h^{(l)}$, thus enabling end-to-end learning of the head roles. During inference, this stochastic process is replaced by a deterministic assignment: a head is designated as a Retrieval Head if its learned expectation $\mathbb{E}[z_h^{(l)}] > 0.5$, and as a Sparse Head otherwise.

**Loss Function and Sparsity Control.** We optimize a distillation loss to align the logits of our hybrid-head student model with those of the full-attention teacher. Given a sequence $X$ partitioned into a prompt $X_{\text{prompt}}$ and a target $X_{\text{target}}$, the teacher encodes $X_{\text{prompt}}$ to produce a shared KV cache. Conditioned on this cache, both models compute logits over the target tokens. Let $\mathbf{y}_T^{(i)}[j]$ and $\mathbf{y}_S^{(i)}[j]$ denote the teacher and student logits, respectively, for the $j$-th target token in the $i$-th sequence of a batch of size $N$. The distillation loss is:

$$\mathcal{L}_{\text{distill}} = \frac{1}{N} \sum_{i=1}^{N} \sum_{j \in X_{\text{target}}} \left\| \mathbf{y}_S^{(i)}[j] - \mathbf{y}_T^{(i)}[j] \right\|_2^2. \tag{6}$$

To enforce a strict sparsity budget on Retrieval Heads, we formulate training as a constrained optimization problem using Lagrangian relaxation. The objective is a min-max problem over the distributional parameters $(\alpha, \beta)$ of the HardKuma selectors and a learnable Lagrange multiplier $\lambda \geq 0$:

$$\min_{\alpha, \beta} \max_{\lambda \geq 0} \mathcal{L}(\alpha, \beta, \lambda) = \mathcal{L}_{\text{distill}} + \lambda \cdot \left( \mathbb{E}[\|\mathbf{z}\|_0] - N_{\text{target}} \right), \tag{7}$$

where the regularizer $\mathbb{E}[\|\mathbf{z}\|_0]$ is the expected $L_0$ norm of the selection variables, which corresponds to the expected number of active Retrieval Heads. The expectation can be expressed in closed form:

$$\mathbb{E}\left[\|\mathbf{z}\|_0\right] = \sum_{l>0,h} \left(1 - F\left(\frac{-p}{q-p}; \alpha_h^{(l)}, \beta_h^{(l)}\right)\right), \tag{8}$$

where $F$ is the CDF function of the Kumaraswam distribution. The detailed derivation of Equation 8 can be found in the Appendix A.3.

During training process, $(\alpha, \beta)$ are optimized via gradient descent to minimize the objective, while $\lambda$ is updated by gradient ascent according to the constraint violation: if the expected number of active heads exceeds $N_{\text{target}}$, $\lambda$ increases to strengthen the penalty; otherwise, it decreases. This adaptive scheme automatically tunes the effective penalty strength, ensuring the desired sparsity without manual hyperparameter search.

## 4 EXPERIMENTS

### 4.1 EXPERIMENT SETTING

**Benchmarks, Models, and Baselines**  We conduct experiments on both efficiency and performance of LycheeDecode. In Section 4.2, we analyze performance under two scenarios: long-context understanding and complex reasoning. For long-context understanding, we benchmark the Llama3-8B and Qwen3-8B models on the LongBench dataset, comparing LycheeDecode against advanced sparse attention methods such as TidalDecode (Yang et al., 2025b), Quest (Tang et al., 2024), DuoAttention (Xiao et al., 2025) and SeerAttention-R (Gao et al., 2025). For complex reasoning, we assess the DeepSeek-R1-Distill-Qwen-7B/Llama-8B models on challenging mathematical reasoning benchmarks, including AIME24 and OlympiadBench. In Section 4.3, we turn to efficiency analysis. Leveraging our custom hybrid-head sparse attention kernels, we conduct a head-to-head comparison with existing sparse attention methods, measuring both end-to-end speedup and kernel-level acceleration.

**Training Setup for LycheeDecode**  To categorize the attention heads, we follow prior work (Xiao et al., 2025), inserting passkeys into the Booksum dataset and calculating a distillation loss through passkey retrieval. In training phase, We trained for 3000 steps on a single NVIDIA A100 80G GPU using a single batch size, which took only a few hours. The HardKuma distribution for each attention head is initialized to a uniform distribution, i.e., parameters $\alpha$ and $\beta$ are both initialized to 1. The critical token budget is set to 30% of the sequence length. For a fair comparison with TidalDecode, the retrieval head budget was set to 32, matching the number of heads that perform full attention in TidalDecode (two full attention layer and two token selection layers, with 8 KV heads each).

### 4.2 PERFORMANCE EVALUATION

#### 4.2.1 LONG CONTEXT UNDERSTANDING

We evaluate the model's ability to understand long contexts on the LongBench (Bai et al., 2024), a benchmark designed to evaluate LLMs on long-context tasks across diverse NLP domains. Following previous work (Yang et al., 2025b), we concentrate on eight tasks that span single/multi-document question answering, summarization, and retrieval: MultiFieldQA (MFQA), NarrativeQA (NrtQA), Qasper (Qasp), 2WikiMQA (2Wiki), HotpotQA (HotQA), QMSum (QMSm), TriviaQA (TrQA), and Passage Retrieval (PRe).

The results, as detailed in Table 1, demonstrate that on the Llama-3-8B-Instruct-Gradient-1048k model, LycheeDecode achieves an average score of 33.07 with 4096 token budget, not only outperforms other sparse attention methods like TidalDecode and Quest but also surpasses the full-attention model in the average score. On the Qwen3-8B model, LycheeDecode outperforms TidalDecode with both 1024 and 4096 token budget, which demonstrates the clear advantage of LycheeDecode's head-level token sharing strategy over the layer-level sharing approach used by TidalDecode. Furthermore, compared to SeerAttention-R, which relies on a trainable gating network, LycheeDecode achieves comparable or slightly superior performance. This demonstrates that our lightweight head

Table 1: Performance comparison on LongBench benchmark. LycheeDecode achieves the best average score in all settings, surpassing other sparse attention methods and full attention models. "∗" indicates double the retrieval head budget. We **bold** the best-performing scores with the second-best underlined.

| Method (Budget) / Task | | MFQA | NrtQA | Qasp | 2Wiki | HotQA | QMSm | TrQA | PRe | Avg. |
|---|---|---|---|---|---|---|---|---|---|---|
| Llama-3-8B-Instruct-Gradient-1048k | | | | | | | | | | |
| Full Attention | | 30.76 | 5.52 | 14.56 | 13.32 | 11.50 | 19.43 | 86.56 | 77.00 | 32.33 |
| Quest | (1024) | 26.21 | 4.08 | 12.19 | 12.61 | 10.75 | 19.56 | 83.47 | 63.84 | 29.09 |
| DuoAttention | (1024) | 19.02 | 7.36 | 8.60 | 9.68 | 8.77 | 17.75 | 41.92 | 13.25 | 15.79 |
| DuoAttention∗ | (1024) | 23.88 | 6.27 | 10.44 | 10.41 | 7.48 | 19.00 | 80.61 | 47.17 | 25.66 |
| TidalDecode | (1024) | 28.57 | **7.63** | 11.11 | 13.56 | 9.82 | **20.37** | 79.78 | 75.17 | 30.75 |
| LycheeDecode | (1024) | 28.28 | 6.12 | **14.89** | **14.42** | 12.81 | 19.05 | 82.69 | 69.92 | 31.02 |
| Quest | (4096) | 28.92 | 3.74 | 13.63 | 12.83 | 12.15 | 19.36 | 85.91 | 72.50 | 31.13 |
| DuoAttention | (4096) | 22.27 | 7.16 | 13.93 | 12.74 | 10.73 | 17.93 | 83.76 | 34.75 | 25.41 |
| DuoAttention∗ | (4096) | 23.74 | 6.63 | 13.80 | 13.67 | 10.40 | 17.93 | 86.03 | 61.00 | 29.15 |
| TidalDecode | (4096) | **30.94** | 6.19 | 13.85 | 14.40 | **13.71** | 19.48 | 86.30 | 78.00 | 32.86 |
| LycheeDecode | (4096) | 30.11 | 5.85 | 14.39 | 12.86 | 12.66 | 19.30 | **86.78** | **82.58** | **33.07** |
| Qwen3-8B | | | | | | | | | | |
| Full Attention | | **25.84** | **3.43** | 10.96 | 11.97 | 11.74 | **20.90** | 90.21 | 89.08 | 33.02 |
| SeerAttention-R | (1024) | 23.91 | 2.97 | 10.28 | 11.88 | 11.28 | 19.04 | 87.50 | 86.79 | 31.71 |
| TidalDecode | (1024) | 21.32 | 2.73 | 9.96 | 10.48 | 9.97 | 19.27 | 80.4 | 83.43 | 29.70 |
| LycheeDecode | (1024) | 24.26 | 3.14 | 10.45 | 11.05 | **12.00** | 19.81 | 86.64 | 91.71 | 32.38 |
| SeerAttention-R | (4096) | 24.85 | 3.30 | **11.15** | 12.42 | 11.35 | 20.61 | 90.19 | 93.17 | 33.38 |
| TidalDecode | (4096) | 23.57 | 2.99 | 10.79 | 11.47 | 11.31 | 20.01 | 88.94 | 85.0 | 31.76 |
| LycheeDecode | (4096) | 24.90 | 3.32 | 10.88 | **12.74** | 11.68 | 20.71 | **90.34** | **93.25** | **33.48** |

identification strategy can effectively capture critical information without the complexity of training and deploying an auxiliary gating network.

Table 2: Performance comparison on math reasoning tasks.

| Method / Task | Gaokao2023En | Minerva | AIME24 | OlympiadBench | Avg. |
|---|---|---|---|---|---|
| DeepSeek-R1-Distill-Llama-8B | | | | | |
| Full Attention | **68.8** | 39.1 | 23.3 | 10.2 | 35.4 |
| TidalDecode | 62.5 | 39.8 | 13.3 | **10.9** | 31.6 |
| TidalDecode w/ Cache Correction | 57.0 | **43.0** | 33.3 | 9.4 | 35.7 |
| LycheeDecode | **68.8** | 40.6 | 26.7 | **10.9** | 36.8 |
| LycheeDecode w/ Cache Correction | **68.8** | 41.4 | **40.0** | **10.9** | **40.3** |
| DeepSeek-R1-Distill-Qwen-7B | | | | | |
| Full Attention | **74.2** | 47.7 | 40.0 | 10.2 | 43.0 |
| TidalDecode | 57.8 | 39.1 | 16.7 | 7.0 | 30.2 |
| TidalDecode w/ Cache Correction | 63.3 | 41.4 | 26.7 | 8.6 | 35.0 |
| LycheeDecode | **74.2** | **48.4** | 43.3 | 10.9 | 44.2 |
| LycheeDecode w/ Cache Correction | 72.7 | 47.7 | **46.7** | **12.5** | **44.9** |

### 4.2.2 COMPLEX REASONING TASK

To evaluate the reasoning capabilities of LycheeDecode, we conduct experiments on four challenging math reasoning benchmarks: Gaokao2023En (Liao et al., 2024), Minerva (Lewkowycz et al., 2022), AIME24 (MAA, 2024), and OlympiadBench (He et al., 2024). We compare our method against Full Attention and TidalDecode on two distilled models from the DeepSeek-R1. In our experimental configuration, the number of tokens for sparse attention calculation is set to half of the sequence length, increasing linearly during decoding. Furthermore, to mitigate the potential accumulation of errors from sparse attention mechanisms, we incorporate a **Cache Correction** strategy (Yang et al., 2025b; Sun et al., 2025). Specifically, after every 32 decoded tokens, we perform

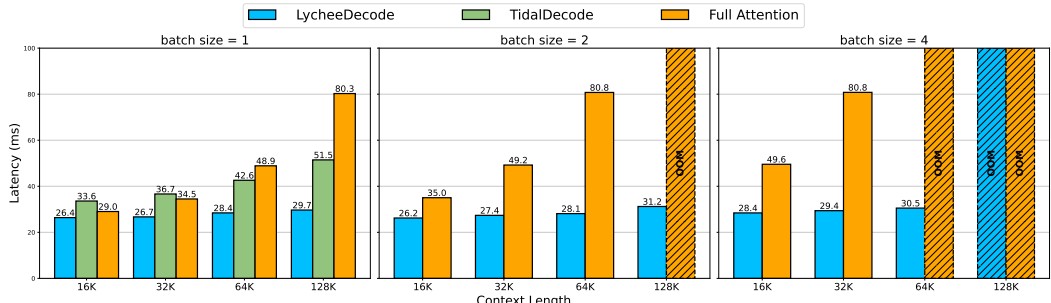

Figure 3: End-to-End Decoding Latency (TPOT) across various context lengths. LycheeDecode and TidalDecode use a fixed 4096 budget. Note that TidalDecode can only support single batch.

a prefill step over these "polluted" tokens using dense attention to reconstruct and update their key-value (KV) representations within the cache.

As demonstrated in Table 2, LycheeDecode outperforms both the TidalDecode and full attention baselines across both models. The introduction of the Cache Correction strategy further enhances the performance of LycheeDecode, solidifying its superiority. We hypothesize that this advantage over the full-attention model stems from our method's ability to capture more diverse attention patterns through head specialization, which allows LycheeDecode to more effectively focus on the key information crucial for the reasoning process while filtering out irrelevant context that may act as noise, leading to a more robust and efficient inference.

### 4.3 EFFICIENCY EVALUATION

#### 4.3.1 END-TO-END SPEEDUP

We evaluate the end-to-end decoding latency of LycheeDecode and compare it against TidalDecode and the full attention baseline across varying context lengths and batch sizes. We adopt TPOT (Time Per Output Token) as the primary evaluation metric. LycheeDecode and TidalDecode use a fixed 4096 token budget. LycheeDecode leverages our efficient hybrid-head block-sparse decoding kernel, combined with auto-tuning to search for the optimal parameter settings in each layer, since different layers contain varying numbers of sparse heads.

As shown in Figure 3, as the context length grows, the latency of the full-attention model increases sharply. TidalDecode exhibits higher latency than full attention at shorter sequence lengths, but surpasses it in longer contexts (>64K). By comparison, LycheeDecode consistently maintains low latency as sequence length increases, achieving up to 2.7× speedup over full attention and 1.73× faster than TidalDecode under a single batch size with 128K context. These results demonstrate that LycheeDecode delivers robust end-to-end acceleration across different settings.

#### 4.3.2 KERNEL-LEVEL SPEEDUP

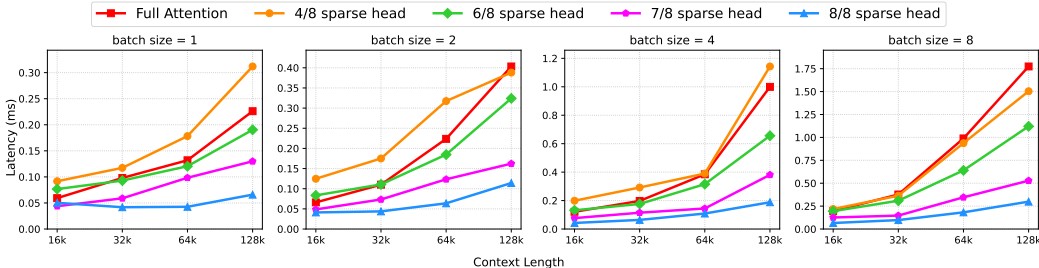

Figure 4: Latency comparison of our hybrid head kernel and the FlashAttention-2 kernel across different sparse head ratios, context lengths, and batch sizes.

This section evaluates our custom hybrid head block-sparse decoding kernel (detailed design is shown in Appendix C). We implement the kernel using TileLang and select FlashAttention-2 (Dao,

2024) as our baseline. Experiments are conducted on single NVIDIA A800 GPU across different context lengths (16K to 128K) and batch sizes (1 to 8). We evaluate several configurations of our kernel, progressively increasing the ratio of sparse heads from 4/8 to 8/8 (out of 8 total key-value heads), with a fixed 90% sparsity ratio applied to the sparse heads and the block size set to 64.

The experimental results clearly validate the efficiency of our custom hybrid-head kernel. As shown in Figure 4, while the configuration with 4/8 sparse heads exhibits latency comparable to or slightly underperforming the dense FlashAttention-2 baseline, all other settings with a higher degree of sparsity consistently and significantly outperform it. This performance advantage becomes particularly pronounced as the input sequence length and batch size increase, which is expected, as the decoding kernel is primarily I/O-bound. When the KV cache size is sufficient to saturate memory bandwidth, the gains are substantial; for instance, at a 128K context length with a batch size of 8, our kernel achieves a peak speedup of up to 7x in the fully sparse (8/8) configuration. This evaluation confirms that our specialized kernel effectively translates the algorithmic gains of the hybrid-head strategy into significant kernel-level acceleration by minimizing redundant computation and memory access, serving as the fundamental enabler for the end-to-end speedups observed in LycheeDecode.

## 4.4 ABLATION STUDY

### 4.4.1 DIFFERENT SPARSITY METHODS

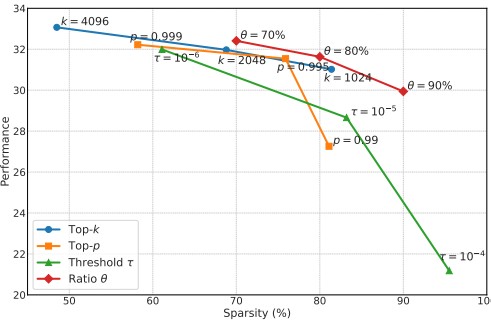

Table 3: Performance comparison of different head identification methods on different datasets. Scores are averaged across eight selected tasks in LongBench.

| Method / Dataset | Passkey Retrieval | HotpotQA |
|---|---|---|
| Direct Optimize | 32.06 | 31.02 |
| Hard Concrete | 32.13 | 30.25 |
| HardKuma (Ours) | 33.07 | 31.11 |

Figure 5: Results of LycheeDecode using different sparse method on the LongBench.

To evaluate the effectiveness of different sparsity strategies, we conduct a comparative analysis of their performance-sparsity trade-offs. We benchmark four distinct families of token selection methods, each with three different configurations: (1) `Top-k`, which retains a fixed-size set of tokens with the highest attention scores; (2) `Top-p`, which adaptively selects the smallest set of tokens whose cumulative attention probability exceeds a predefined threshold $p$; (3) `Threshold`, which preserves all tokens with attention scores surpassing a specific value; and (4) `Ratio`, which selects a set of top tokens using a budget proportional to the sequence length, designed to increase gradually during the generation process.

For each configuration, we measure two key metrics: (1) Performance, quantified by the average F1 score on the LongBench benchmark, and (2) Sparsity, defined as the percentage of critical tokens identified by sparse heads to the total sequence length during inference.

The experimental results are shown in Figure 5. More details can be found in Appendix E.2. Increasing sparsity leads to a decline in model performance. This is expected, as higher sparsity reduces the amount of contextual information available. `Top-p` and `Ratio` perform robustly under low sparsity, sometimes even surpassing `Top-k` with comparable token budgets. However, their performance drops sharply under extreme sparsity. Notably, at equivalent sparsity levels, the `Ratio` method generally achieves the best performance. We hypothesize that training with a fixed-sparsity objective endows the model with a general robustness to sparsity, which in turn allows it to effectively handle the dynamic adjustments made by the `Ratio` method during inference.

### 4.4.2 IDENTIFICATION METHODS & DATASET

To evaluate the advantages of our HardKuma distribution for identifying attention heads, we compare it with the direct optimization baseline from Xiao et al. (2025) and the HardConcrete distribution used by Bhaskar et al. (2025). The head identification process is performed on two distinct datasets: the previously mentioned Passkey Retrieval task and HotpotQA, which challenges the model to perform multi-hop reasoning over long contexts. For HotpotQA, the distillation loss is calculated based on the logits of the answer tokens. Crucially, we filter out questions that can be answered without relying on the provided context, thereby ensuring that the identification process specifically rewards heads capable of complex, long-range information integration. The specialized models are then evaluated on the LongBench benchmark with a fixed 4096 token budget.

As shown in Table 3, the HardKuma distribution achieves the best overall performance, outperforming both the direct optimization baseline and HardConcrete distribution and demonstrating its superior ability to identify head type. Its score is slightly lower on the HotpotQA dataset, which we hypothesize this is because its answers are relatively short; calculating the loss over a small number of tokens can lead to a higher variance in the gradient estimate, making it difficult to accurately guide the specialization of attention heads. We leave the optimization of tasks where the supervision signal is sparse for future work. Refer to Appendix A for more discussion of theoretical advantages.

## 5 CONCLUSION

We introduce LycheeDecode, a framework that speeds up long-context LLMs by specializing attention heads for different roles, enhancing efficiency while maintaining performance. This head specialization is enabled by the HardKuma distribution and a custom TileLang kernel, delivering significant end-to-end speedups. Our work highlights that treating attention heads as functionally specialized units, rather than a monolithic block, is a powerful and promising direction for LLMs.

## 6 ACKNOWLEDGMENTS

This work was supported by the National Natural Science Foundation of China (Grant No. 62406088, 62422603), Guangdong Basic and Applied Basic Research Foundation (Grant No. 2025A1515011376), Guangdong Basic and Applied Basic Research Foundation (Grant No. 2024B0101050003). Baotian Hu is the corresponding author.

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

# A  HARDKUMA DISTRIBUTION

## A.1  KUMARASWAMY DISTRIBUTION

The Kumaraswamy (Kuma) distribution is a continuous probability distribution defined on the interval (0,1). It is similar to the Beta distribution, but its probability density function (PDF) and cumulative distribution function (CDF) are simpler and have closed form expressions.

The PDF of the Kumaraswamy distribution is given by:

$$f(x; \alpha, \beta) = \alpha \beta x^{\alpha-1}(1 - x^\alpha)^{\beta-1}, \tag{9}$$

where $x \in (0, 1)$, $\alpha$ and $\beta$ are positive shape parameters that control the distribution's shape.

The shape of the distribution can be unimodal, uniantimodal, increasing, decreasing, or constant, depending on the values of $\alpha$ and $\beta$.

The CDP of Kumaraswamy distribution can be defined as:

$$\begin{aligned}
F(x; \alpha, \beta) &= \int_0^x f(\xi; \alpha, \beta) d\xi \\
&= \int_0^x \alpha \beta \xi^{\alpha-1}(1 - \xi^\alpha)^{\beta-1} d\xi
\end{aligned} \tag{10}$$

Let $u = 1 - \xi^\alpha$, then the differential is $du = -\alpha \xi^{\alpha-1} d\xi$. We also need to change the limits of integration: when $\xi = 0$, $u = 1$, and when $\xi = x$, $u = 1 - x^\alpha$. Substituting these into the integral gives:

$$\begin{aligned}
F(x; \alpha, \beta) &= -\beta \int_1^{1-x^\alpha} u^{\beta-1} du \\
&= -\beta \left[ \frac{u^\beta}{\beta} \right]_1^{1-x^\alpha} \\
&= 1 - (1 - x^\alpha)^\beta
\end{aligned} \tag{11}$$

The PDF and CDF of the Kuma distribution with different parameters are shown in Figure 6.

## A.2  HARDKUMA DISTRIBUTION

The HardKuma distribution is a modification of the Kumaraswamy distribution, engineered to create a random variable on the closed interval that exhibits both continuous and discrete behavior. It achieves this by having non-zero probability masses at the endpoints $0$ and $1$, while maintaining a continuous density over the open interval $(0, 1)$. This makes it particularly useful for applications like generating differentiable binary masks in machine learning.

The distribution is constructed as follows. Let $X$ be a random variable following the Kumaraswamy distribution, i.e., $X \sim \text{Kuma}(\alpha, \beta)$. We define an intermediate *stretched* variable $T$ by linearly transforming $X$ to a wider interval $(p, q)$, where $p < 0$ and $q > 1$ are fixed hyperparameters:

$$T = p + (q - p)X \tag{12}$$

The HardKuma random variable, which we denote as $Z$, is then obtained by applying a hard-sigmoid rectifier function to $T$:

$$Z = \min(1, \max(0, T)) \tag{13}$$

A variable $Z$ constructed this way is said to follow the HardKuma distribution, i.e., $Z \sim \text{HardKuma}(\alpha, \beta)$.

The key feature of this construction is that the discrete probabilities for $Z = 0$ and $Z = 1$ can be computed in closed form, thanks to the tractable CDF of the underlying Kumaraswamy distribution.

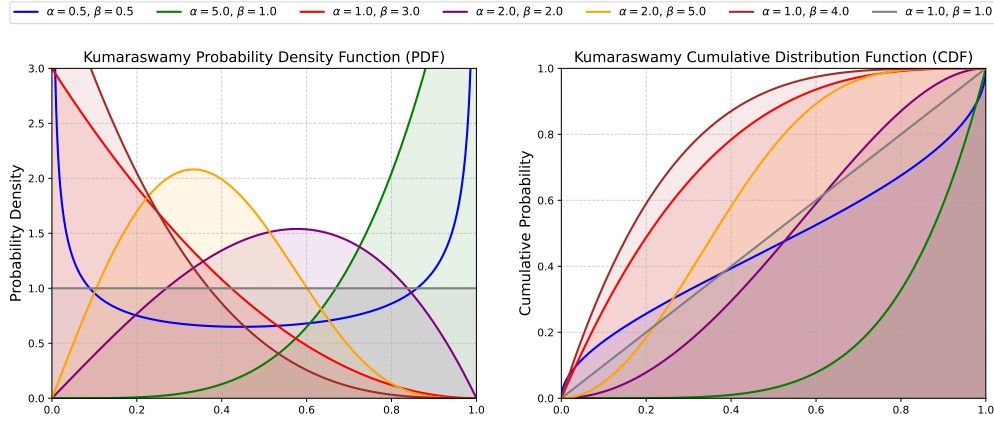

Figure 6: PDF and CDF of Kuma distribution with different parameters.

The probability of sampling exactly $0$ is the probability that the stretched variable $T$ is less than or equal to $0$:

$$
\begin{aligned}
P(Z = 0) &= P(T \leq 0) \\
&= P(p + (q - p)X \leq 0) \\
&= P\left(X \leq \frac{-p}{q - p}\right) \\
&= F\left(\frac{-p}{q - p}; \alpha, \beta\right)
\end{aligned}
\tag{14}
$$

Similarly, the probability of sampling exactly $1$ is the probability that $T$ is greater than or equal to $1$:

$$
\begin{aligned}
P(Z = 1) &= P(T \geq 1) \\
&= 1 - P(T < 1) \\
&= 1 - P\left(X < \frac{1 - p}{q - p}\right) \\
&= 1 - F\left(\frac{1 - p}{q - p}; \alpha, \beta\right)
\end{aligned}
\tag{15}
$$

The remaining probability mass, $1 - P(Z = 0) - P(Z = 1)$, is distributed continuously over the interval $(0, 1)$. This mixed discrete-continuous nature allows the HardKuma distribution to model binary selections in a way that is amenable to gradient-based optimization.

### A.3 EXPECTED $L_0$ NORM OF HARDKUMA

A primary application of the HardKuma distribution is to create sparse, differentiable masks. This involves generating a vector of random variables $\mathbf{Z} = (Z_1, \ldots, Z_n)$, where each $Z_i$ is drawn independently from a HardKuma distribution, $Z_i \sim \text{HardKuma}(\alpha_i, \beta_i)$. The sparsity of such a vector is measured by its $L_0$ norm $\|\mathbf{Z}\|_0$, which counts the number of non-zero elements.

A key result, which makes this distribution practical for optimization, is that the expected value of the $L_0$ norm has a tractable, closed-form expression. We can derive it as follows.

First, we express the $L_0$ norm using the indicator function $\mathbb{I}[\cdot]$:

$$
\|\mathbf{Z}\|_0 = \sum_{i=1}^{n} \mathbb{I}[Z_i \neq 0]
\tag{16}
$$

By the linearity of expectation, the expectation of the sum is the sum of the expectations:

$$\mathbb{E}[\|\mathbf{Z}\|_0] = \mathbb{E}\left[\sum_{i=1}^{n} \mathbb{I}[Z_i \neq 0]\right] = \sum_{i=1}^{n} \mathbb{E}[\mathbb{I}[Z_i \neq 0]] \tag{17}$$

The expectation of an indicator function is simply the probability of the event it indicates:

$$\mathbb{E}[\mathbb{I}[Z_i \neq 0]] = P(Z_i \neq 0) \tag{18}$$

Using the complement rule, the probability of being non-zero is one minus the probability of being zero:

$$P(Z_i \neq 0) = 1 - P(Z_i = 0) \tag{19}$$

Combining these steps and Equation 14 , we arrive at the final expression for the expected $L_0$ norm:

$$
\begin{aligned}
\mathbb{E}[\|\mathbf{Z}\|_0] &= \sum_{i=1}^{n}(1 - P(Z_i = 0)) \\
&= \sum_{i=1}^{n}\left(1 - F\left(\frac{-p}{q-p}; \alpha_i, \beta_i\right)\right)
\end{aligned}
\tag{20}
$$

## B  ALGORITHM PSEUDOCODE

The complete procedure of LycheeDecode is shown in Algorithm 1. In each layer, the Key-Value (KV) cache is first updated with the key and value vectors of the current token. The algorithm then processes each attention head according to its designated type: Retrieval Heads perform a full attention operation over the entire KV cache to identify and select a new set of critical tokens. Conversely, Sparse Heads perform a more efficient computation, calculating attention only on the sparse subset of tokens provided by the preceding layer. Following the attention step, the outputs from all heads are concatenated and passed through a feed-forward network to produce the hidden state for the subsequent layer. This entire procedure is repeated until the final logits are produced by the model's output layer.

---

**Algorithm 1** LycheeDecode

---

1: **Input:** Initial hidden state $x^{(0)}$, KV cache $\mathcal{C}$, selected token set $\{\mathcal{S}_h\}_{h=0}^{H-1}$, token budget $k$
2: **Output:** Logits
3: **for** layer $l = 0, 1, \ldots, L-1$ **do**
4:     $q, k, v \leftarrow x^{(l)} W_Q, x^{(l)} W_K, x^{(l)} W_V$
5:     $\mathcal{C}^{(l)}.\text{append}(k, v)$
6:     $K, V \leftarrow \mathcal{C}^{(l)}.\text{key}, \mathcal{C}^{(l)}.\text{value}$
7:     **for** head $h = 0, 1, \ldots, H-1$ **do**
8:         **if** $l == 0$ **or** $h \in \mathcal{H}_R^{(l)}$ **then**         ▷ Retrieval Head
9:             $A_h \leftarrow \text{softmax}\left(q_h K_h^T / \sqrt{d}\right)$
10:            $\mathcal{S}_h \leftarrow \text{argTopK}(A_h, k)$         ▷ Select $k$ critical tokens
11:            $o_h \leftarrow A_h V_h$
12:         **else**         ▷ Sparse Head
13:            $o_h \leftarrow \text{softmax}\left(q_h (K_h[\mathcal{S}_h])^T / \sqrt{d}\right) V_h[\mathcal{S}_h]$
14:         **end if**
15:     **end for**
16:     $o \leftarrow \text{Concat}(o_0, o_1, \ldots, o_{H-1}) W_O$
17:     $x^{(l+1)} \leftarrow \text{FFN}(o)$
18: **end for**
19: $\text{logits} \leftarrow \text{lm\_head}(x^{(L-1)})$
20: **return** logits

---

## C  KERNEL DESIGN

---

**Algorithm 2** Hybrid-head Block-Sparse Decoding

---

 1: **Input:** Query $q$, Key $K$, Value $V$, block indices $I$
 2: **Output:** Attention output $O$
 3: **for** Grid indexed $(b, s)$ by $(\text{batch\_size}, \text{num\_split})$ in parallel **do**
 4:     Calculate head_id $h$ and head-wise split_id $s_h$ base on the sparse head index and split_id $s$
 5:     Load corresponding query block $q_{b,h}$ in a GQA group into shared memory
 6:     $o_{partial} \leftarrow 0, m_{partial} \leftarrow -\infty, l_{partial} \leftarrow 0$                    ▷ Initialize accumulators
 7:     **for** each block index $i \in I$ within the current split **do**
 8:         Load corresponding key block $K_i$ and value block $V_i$ into shared memory
 9:         $S_i = q_{b,h} \cdot K_i^T$                       ▷ Compute score matrix via GEMM operation
10:         Update $o_{partial}, m_{partial}, l_{partial}$ with $S_i, V_i$ using online softmax algorithm
11:     **end for**
12:     $O_{partial}[b, h, s_h] \leftarrow o_{partial}/l_{partial}$                         ▷ Store partial output
13:     $L_{partial}[b, h, s_h] \leftarrow \log(l_{partial}) + m_{partial}$              ▷ Store partial log-sum-exp
14: **end for**
15: Combine$(L_{partial}, O_{partial}, O)$                              ▷ Combine different splits
16: **return** $O$

---

A critical challenge in designing an efficient hybrid-head attention kernel is the inherent workload imbalance between the different head types. Retrieval heads, which must process the entire Key-Value cache, represent a substantial computational load. In contrast, sparse heads operate on only a small, pre-selected subset of blocks, demanding significantly fewer resources. A naive scheduling approach that allocates an equal number of computational resources, such as GPU thread blocks, to each head would result in a severe performance bottleneck. Threads assigned to sparse heads would complete their tasks rapidly and remain idle, while the threads dedicated to full-attention heads would dictate the critical path, leading to gross underutilization of the GPU's parallel architecture.

To overcome this, we implement a workload-pooling strategy in our hybrid-head sparse decoding kernel that decouples resource allocation from individual heads. Instead of assigning work on a per-head basis, we first aggregate the complete set of block computations required by all heads (both full and sparse) into a single, unified pool of work for each batch item. This aggregated workload is then partitioned into numerous smaller, uniform work units, which we term *splits*. These splits are subsequently distributed homogeneously among the available GPU thread blocks for execution. By aggregating the heterogeneous computations before partitioning, this approach ensures that every thread block receives a workload of roughly equivalent size, maximizing hardware utilization and minimizing overall execution latency. See Algorithm 2 for detailed pseudo code.

## D  VISUALIZATION OF TRAINING PROCESS

To demonstrate the effectiveness of our proposed head identification strategy in bridging the train-inference gap, we visualize the training dynamics of LycheeDecode alongside the baseline DuoAttention (Xiao et al., 2025). We conducted the comparison on the Llama-3-8B-Instruct-1048k model, training both for 1000 steps with an identical learning rate of 0.01. Figure 7 presents the evolution of the probability that a specific attention head is identified as a "Retrieval Head" during training.

For DuoAttention, the heatmap values represent the continuous gating variables. As observed, DuoAttention exhibits noticeable "grey" areas (values hovering between 0.4 and 0.6) at step 1000. This indicates that a simple continuous relaxation often fails to push parameters to the binary extremes. Consequently, rounding these ambiguous values to 0 or 1 during inference introduces a substantial train-inference discrepancy, potentially degrading performance.

For LycheeDecode, the heatmap values represent the expected value $E[z_h^{(l)}]$ of the HardKuma distribution. In contrast to the DuoAttention, LycheeDecode demonstrates a more decisive polarization. The values quickly converge to either 0 (Sparse Head) or 1 (Retrieval Head), resulting in a clear "blue-and-red" pattern. This confirms that the HardKuma distribution effectively forces the model

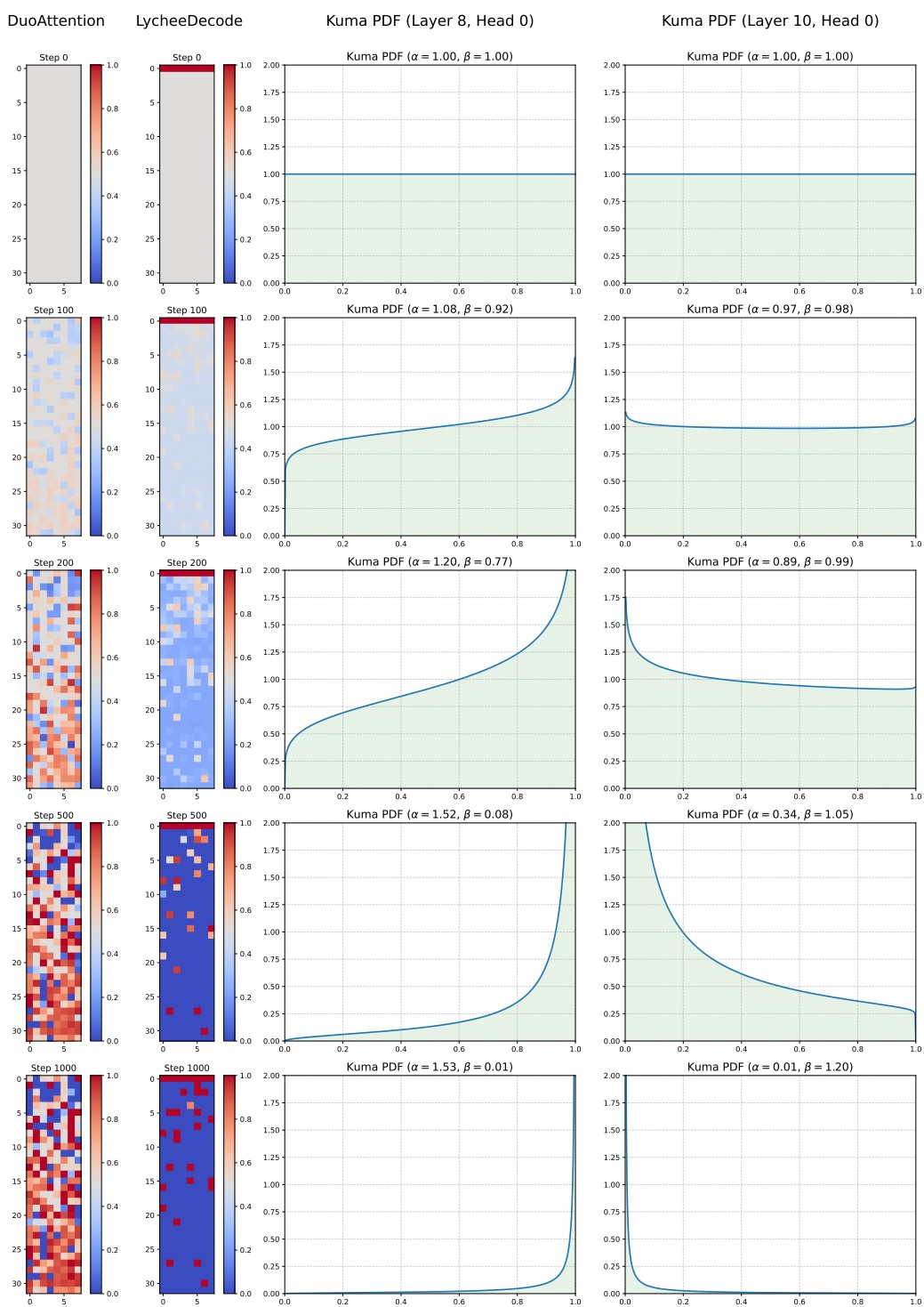

Figure 7: Visualization of head specialization dynamics on Llama-3-8B-Instruct-1048k. **Left & Middle (Heatmaps)**: The probability of each head being identified as a Retrieval Head across training steps for DuoAttention (left) and LycheeDecode (middle). **Right (PDFs)**: Evolution of the LycheeDecode Kuma distribution PDFs for specific heads at steps 0, 100, 200, 500, and 1000, showing how probability mass effectively concentrates at the boundaries.

to make discrete decisions during the training phase itself, thereby minimizing the consistency gap between training and inference.

The two rightmost columns of Figure 7 provide a microscopic view of this process by plotting the Probability Density Functions (PDFs) of the Kuma distribution for two representative heads (Layer 8 Head 0 and Layer 10 Head 0) at specific training steps. Initially uniform at Step 0, the distributions undergo a dramatic transformation. For the head specializing as a Retrieval Head, the probability mass shifts almost entirely to the right, while for the Sparse Head, it collapses to the left. This visualization corroborates that our optimization objective successfully shapes the underlying distribution to be near-binary.

# E  MORE EXPERIMENT RESULTS

## E.1  RULER BENCHMARK

To assess the ability to comprehend longer contexts, we employ the RULER benchmark (Hsieh et al., 2024), a synthetic benchmark designed for a more thorough evaluation of long-context language models beyond simple retrieval tasks. RULER expands on the needle-in-a-haystack (NIAH) test by including more complex tasks like multihop tracing and aggregation, offering configurable sequence lengths and task difficulties. For our evaluation, we selected tasks including *niah_single1*, *niah_multikey1*, *niah_multivalue*, *niah_multiquery*, *vt*, *fwe*, *qa1*, and *qa2* to test a wide range of long-context understanding capabilities. We configure LycheeDecode with a fixed budget of 4096 tokens and compare it to the full-attention Llama3-8B-Instruct-Gradient-1048k model.

The experimental results are shown in Table 4. As indicated, in shorter context scenarios, the performance of our method is highly competitive with the full attention model. For instance, at 8k context length, our approach achieves an average score of 62.79, closely approaching the full-attention model's score of 63.30. As the context length increases, the performance of LycheeDecode decreases slightly. This performance degradation is an acceptable trade-off, given that our method operates on a fixed and significantly smaller 4096 token budget.

Table 4: Performance comparison of LycheeDecode and full attention model on RULER benchmark. LycheeDecode uses a fixed budget of 4096.

| Context / Task | single | multikey | multivalue | multiquery | vt | fwe | qa1 | qa2 | Avg. |
|---|---|---|---|---|---|---|---|---|---|
| Full Attention | | | | | | | | | |
| 4k | 100.0 | 89.6 | 87.8 | 79.2 | 17.4 | 0.1 | 79.8 | 56.4 | 63.7 |
| 8k | 100.0 | 95.0 | 90.3 | 70.0 | 19.4 | 0.4 | 75.0 | 56.4 | 63.3 |
| 16k | 100.0 | 93.0 | 95.7 | 81.0 | 19.8 | 0.0 | 74.2 | 53.4 | 64.6 |
| 32k | 99.2 | 97.4 | 96.5 | 81.9 | 19.8 | 0.0 | 70.6 | 51.6 | 65.9 |
| 64k | 99.4 | 98.4 | 96.8 | 93.7 | 19.8 | 0.0 | 70.4 | 47.6 | 65.8 |
| LycheeDecode | | | | | | | | | |
| 4k | 100.0 | 89.4 | 88.4 | 78.9 | 17.3 | 0.1 | 80.0 | 56.2 | 63.7 |
| 8k | 100.0 | 94.4 | 90.6 | 65.9 | 19.4 | 0.4 | 75.4 | 56.2 | 62.8 |
| 16k | 100.0 | 81.8 | 96.3 | 68.7 | 19.6 | 0.0 | 71.0 | 53.4 | 61.4 |
| 32k | 97.8 | 82.0 | 94.9 | 65.1 | 19.8 | 0.0 | 66.2 | 49.6 | 59.4 |
| 64k | 99.6 | 73.2 | 90.3 | 81.7 | 19.9 | 0.0 | 63.4 | 44.4 | 59.0 |

## E.2  DETAILED RESULTS OF DIFFERENT SPARSE METHODS

This section provides a detailed breakdown of the results from the ablation study on different sparsity methods, as discussed in Section 4.4.1 and visualized in Figure 5. Table 5 presents the performance results of LycheeDecode on the LongBench benchmark when configured with different token selection methods, including `Top-k`, `Top-p`, `Threshold`, and `Ratio`, each with varying parameters. Complementing this, Table 6 quantifies the sparsity level (as a percentage of critical tokens selected) for each corresponding strategy and setting.

Table 5: Performance comparison of LycheeDecode using different sparse strategies on LongBench.

| Method / Task | MFQA | NrtQA | Qasp | 2Wiki | HotQA | QMSm | TrQA | PRe | Avg. |
|---|---|---|---|---|---|---|---|---|---|
| Top-$k_{k=1024}$ | 28.28 | 6.12 | 14.89 | 14.42 | 12.81 | 19.05 | 82.69 | 69.92 | 31.02 |
| Top-$k_{k=2048}$ | 28.13 | 5.78 | 14.72 | 12.76 | 11.82 | 19.14 | 84.98 | 78.42 | 31.97 |
| Top-$k_{k=4096}$ | 30.11 | 5.85 | 14.39 | 12.86 | 12.66 | 19.30 | 86.78 | 82.58 | 33.07 |
| Top-$p_{p=0.99}$ | 30.01 | 11.05 | 12.65 | 13.48 | 12.91 | 21.05 | 74.27 | 42.67 | 27.26 |
| Top-$p_{p=0.995}$ | 33.42 | 9.47 | 13.84 | 15.55 | 13.70 | 20.12 | 80.93 | 65.25 | 31.54 |
| Top-$p_{p=0.999}$ | 31.02 | 6.30 | 13.77 | 14.00 | 12.25 | 19.99 | 85.77 | 74.67 | 32.22 |
| Threshold$_{\tau=10^{-4}}$ | 25.22 | 8.17 | 11.63 | 13.78 | 10.86 | 19.75 | 60.08 | 19.93 | 21.18 |
| Threshold$_{\tau=10^{-5}}$ | 28.64 | 6.41 | 14.91 | 15.02 | 13.48 | 19.31 | 78.16 | 53.36 | 28.66 |
| Threshold$_{\tau=10^{-6}}$ | 29.73 | 6.74 | 14.00 | 13.75 | 11.49 | 19.71 | 83.48 | 77.08 | 31.99 |
| Ratio$_{\theta=70\%}$ | 26.68 | 6.54 | 15.58 | 14.10 | 12.95 | 18.91 | 83.82 | 80.67 | 32.41 |
| Ratio$_{\theta=80\%}$ | 26.65 | 6.88 | 13.59 | 15.70 | 11.97 | 19.19 | 81.94 | 77.17 | 31.63 |
| Ratio$_{\theta=90\%}$ | 27.96 | 6.61 | 11.91 | 15.01 | 13.33 | 18.95 | 80.41 | 65.33 | 29.94 |

Table 6: Sparsity (%) of LycheeDecode under different settings on LongBench benchmark.

| Method / Task | MFQA | NrtQA | Qasp | 2Wiki | HotQA | QMSm | TrQA | PRe | Avg. |
|---|---|---|---|---|---|---|---|---|---|
| Top-$k_{k=1024}$ | 87.94 | 92.24 | 67.70 | 71.46 | 82.16 | 88.66 | 86.88 | 74.46 | 81.4 |
| Top-$k_{k=2048}$ | 79.17 | 86.60 | 44.54 | 54.16 | 69.96 | 80.41 | 77.57 | 58.26 | 68.8 |
| Top-$k_{k=4096}$ | 61.91 | 75.32 | 14.21 | 30.80 | 51.62 | 63.92 | 60.12 | 30.17 | 48.5 |
| Top-$p_{p=0.99}$ | 81.81 | 84.59 | 76.34 | 79.85 | 82.12 | 85.78 | 79.71 | 79.17 | 81.1 |
| Top-$p_{p=0.995}$ | 76.22 | 78.50 | 70.42 | 75.06 | 76.67 | 81.50 | 74.99 | 73.80 | 75.9 |
| Top-$p_{p=0.999}$ | 59.67 | 61.02 | 53.66 | 58.40 | 58.32 | 60.76 | 57.29 | 56.37 | 58.2 |
| Threshold$_{\tau=10^{-4}}$ | 96.95 | 98.25 | 91.69 | 93.33 | 95.97 | 97.77 | 96.44 | 93.92 | 95.5 |
| Threshold$_{\tau=10^{-5}}$ | 86.20 | 90.38 | 73.22 | 78.60 | 85.05 | 88.44 | 85.12 | 78.87 | 83.2 |
| Threshold$_{\tau=10^{-6}}$ | 65.36 | 72.07 | 48.04 | 55.11 | 62.89 | 68.05 | 62.92 | 54.21 | 61.1 |
| Ratio$_{\theta=90\%}$ | 90.00 | 90.00 | 90.00 | 90.00 | 90.00 | 90.00 | 90.00 | 90.00 | 90.0 |
| Ratio$_{\theta=80\%}$ | 80.00 | 80.00 | 80.00 | 80.00 | 80.00 | 80.00 | 80.00 | 80.00 | 80.0 |
| Ratio$_{\theta=70\%}$ | 70.00 | 70.00 | 70.00 | 70.00 | 70.00 | 70.00 | 70.00 | 70.00 | 70.0 |

### E.3 MORE CASES

In this section, we provide additional examples to illustrate the behavioral differences among various attention heads. We use prompts that require simple logical reasoning. For each attention head, we calculate the attention scores of the final answer token with respect to all previous tokens and identify the top-$k$ crucial tokens with the highest scores. Subsequently, we compute the overlap rate of these crucial tokens for each attention head with those of the corresponding head in the adjacent layer. The results are presented in Figures 10, Figure 11, Figure 12 and Figure 13.

### E.4 ABLATION STUDY

To investigate the trade-offs between model performance and inference efficiency, we conducted an ablation study using the Llama3-8B-Instruct-Gradient-1048k model. We evaluated the generative quality based on the average score across the LongBench benchmark, while efficiency was quantified by the end-to-end decoding speedup (measured via Time Per Output Token, TPOT) relative to the Full Attention baseline. In this experiment, we explored a range of sparsity configurations by varying two key hyperparameters: the critical token budget, which was set to 1024, 2048, and 4096 tokens, and the ratio of retrieval heads, which was tested at 12.5%, 25.0%, and 50.0% of the total attention heads.

As illustrated in Figure 8, the results demonstrate a clear trade-off between performance and efficiency. We observe that increasing the token budget from 1024 to 4096 consistently enhances the

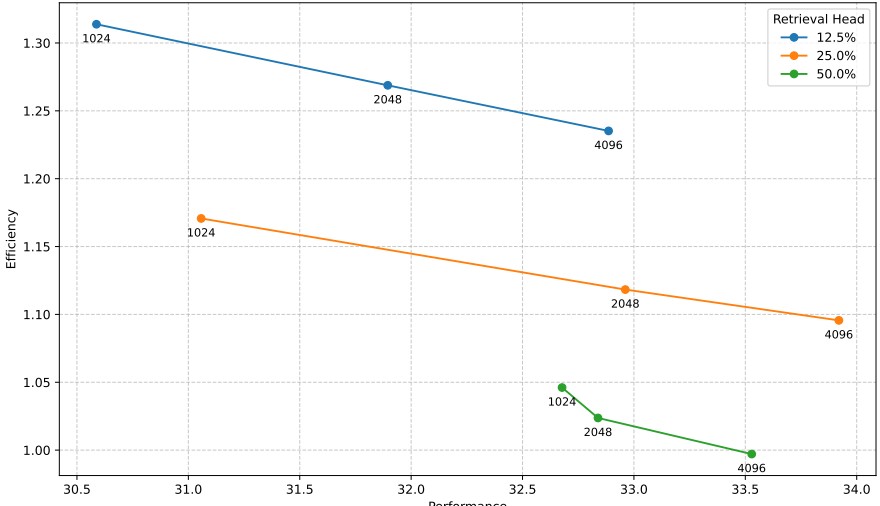

Figure 8: Performance and efficiency trade-offs under different token budgets and retrieval head budgets.

LongBench average score across settings by retaining more context, though this naturally leads to a decrease in decoding speedup. Regarding the proportion of retrieval heads, a smaller ratio yields the highest speedup by minimizing computational overhead. However, in terms of performance, simply maximizing the number of retrieval heads does not always lead to the best results. Notably, with larger token budgets (2048 and 4096), the 25.0% configuration outperforms the 50.0% setting. We hypothesize that this is due to the noise-filtering property of LycheeDecode. An excessive proportion of retrieval heads may introduce irrelevant context, whereas a balanced configuration allows sparse heads to effectively focus on the most critical information.

### E.5 ATTENTION VISUALIZATION

To analyze the behavior of Sparse Heads and investigate how they handle noisy context compared to full-attention, we conducted a case study using a logical reasoning prompt with irrelevant distractor text (Figure 9). We computed the attention weights of the final answer token with respect to the entire preceding context. We calculated the average attention scores across all Retrieval Heads (which execute full attention) and compared them against the average scores across all Sparse Heads (which execute sparse attention). This comparison allows us to directly observe the impact of the sparsity mechanism on the attention distribution.

The comparative visualization is presented in Figure 9. As shown in Figure 9(a), the Retrieval Heads display a diffused attention pattern characteristic of full attention, where significant attention mass is allocated to irrelevant distractor tokens (e.g., "West", "the"). In contrast, Figure 9(b) demonstrates that the Sparse Heads effectively eliminate this noise. Since Sparse Heads constitute the majority of the model's computation, this "denoising" effect explains the counter-intuitive finding that LycheeDecode can outperform the full-attention baseline, as it filters out interference that would otherwise distract the model.

### F IMPLEMENTATION DETAILS

We provide additional experimental details to ensure the reproducibility of our results. For the training phase, we set the learning rate to 0.01. The stretching interval $(p, q)$ for the HardKuma distribution was set to $(-0.1, 1.1)$. For the Passkey Retrieval dataset, we follow the setup of Xiao et al. (2025) by inserting ten 32-word passkeys into the BookSum dataset, with the prompt length sampled from a range of 1k to 10k tokens. For the HotpotQA dataset, we filter out questions that could be answered without requiring the provided context. The prompt length for HotpotQA was

(a) Full Attention

Question: John's father has four children: North, South, and East. What is the name of the fourth child? <think>

These are the cardinal directions. We have North, South, and East. To complete the compass rose, the remaining

direction is West. It logically follows the set of four points. So the final answer is West. Wait, I am overlooking the

introduction. The man is identified as "John's father". Therefore, John is one of the children. The direction pattern is

a distraction. </think> So the final answer is **John**

(b) Sparse Attention

Question: John's father has four children: North, South, and East. What is the name of the fourth child? <think>

These are the cardinal directions. We have North, South, and East. To complete the compass rose, the remaining

direction is West. It logically follows the set of four points. So the final answer is West. Wait, I am overlooking the

introduction. The man is identified as "John's father". Therefore, John is one of the children. The direction pattern is

a distraction. </think> So the final answer is **John**

Figure 9: Visualization of attention scores for the final answer token in a noisy reasoning context. (a) Full Attention (Retrieval Heads) assigns significant attention weight to the irrelevant distraction text (e.g. "West", "the"), indicating susceptibility to noise. (b) Sparse Heads successfully filter out these distractions. By computing attention only on the propagated critical tokens, the Sparse Heads concentrate their focus solely on the relevant reasoning path, effectively denoising the context.

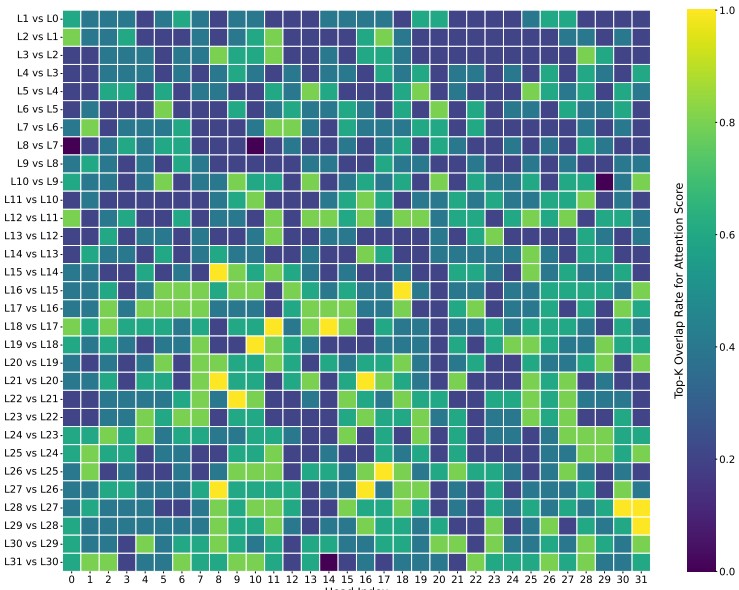

Figure 10: Overlap rate of top-$k$ attention scores between corresponding heads in adjacent layers. The heatmap illustrates the functional diversity among attention heads. We show the overlap rate ($k = 5$) for the prompt: *Please directly output the final answer based on the given question. Question: There are only two kinds of fruit in a box: apples and bananas. All apples are sour, and all bananas are sweet. I took a fruit from the box and tasted it. It was sweet. What is this fruit? Answer:*, and Llama-3 outputs *banana*.

sampled from a range of 1k to 20k tokens. During inference, we preprocess the model by reordering the output channels of the Query, Key, and Value projection weights according to the attention head assignments, so as to ensure that the retrieval head and the sparse head are grouped into two different continuous clusters. For Grouped Query Attention (GQA) models, we reduce the dimension of the Q heads to match that of the KV heads by applying average pooling, which allows for the calculation of the highest-scoring token set for each kv head. For all experiments, we employ the greedy decoding strategy.

## G  LIMITATION

Although LycheeDecode demonstrates a significant step towards efficient long-context LLM inference, we acknowledge several limitations that present valuable avenues for future research. Currently, we allocate a fixed budget for each sparse head. However, recent work (Feng et al., 2024) suggests that dynamically allocating the budget among attention heads can lead to better performance. Additionally, while we have achieved considerable speedup, our method is not yet integrated with highly optimized inference serving frameworks like vLLM (Kwon et al., 2023), which is left for future work.

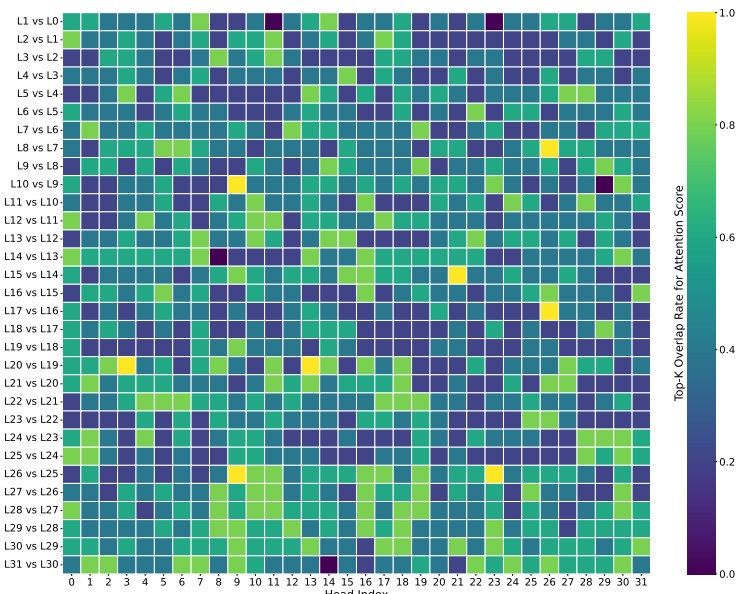

Figure 11: Overlap rate of top-$k$ attention scores between corresponding heads in adjacent layers. The heatmap illustrates the functional diversity among attention heads. We show the overlap rate ($k = 5$) for the prompt: *Please directly output the final answer based on the given question. Question: If you walk 10 meters north from a starting point, then 10 meters east, and finally 10 meters west, what direction are you from the original position? Answer:*, and Llama-3 outputs *north*.

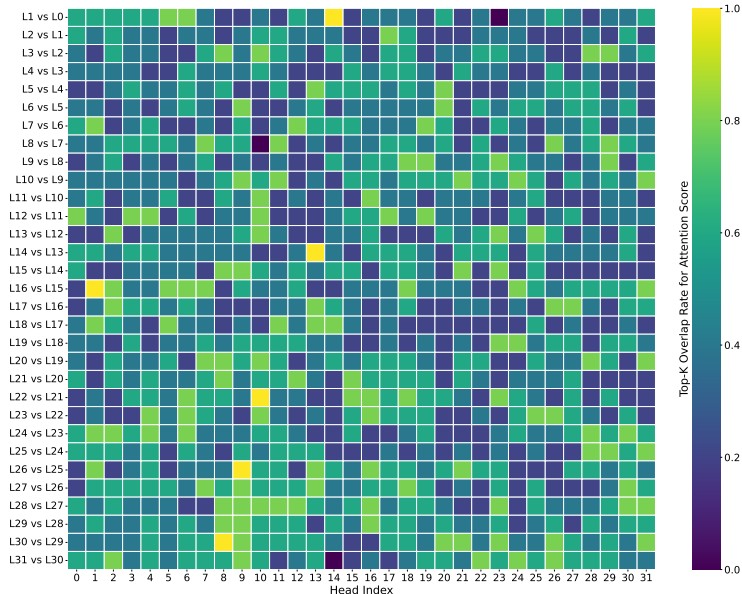

Figure 12: Overlap rate of top-$k$ attention scores between corresponding heads in adjacent layers. The heatmap illustrates the functional diversity among attention heads. We show the overlap rate ($k = 5$) for the prompt: *Please directly output the final answer based on the given question. Question: You start facing east. You turn left 90 degrees, then turn right 180 degrees, and finally turn left 90 degrees. What direction are you facing now? Answer:*, and Llama-3 outputs *east*.

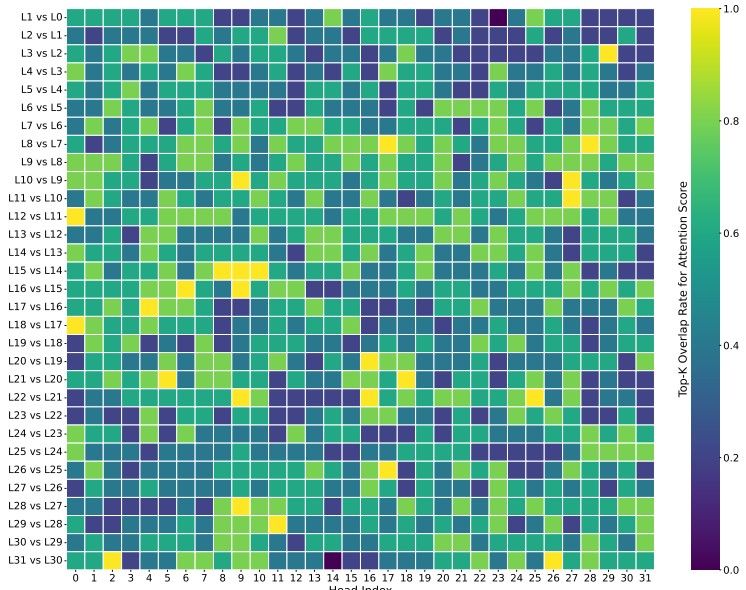

Figure 13: Overlap rate of top-$k$ attention scores between corresponding heads in adjacent layers. The heatmap illustrates the functional diversity among attention heads. We show the overlap rate ($k = 5$) for the prompt: *Please directly output the final answer based on the given question. Question: If two days ago was Monday, what day is tomorrow? Answer:*, and Llama-3 outputs *Thursday*.

