# OpenReview forum: "LycheeDecode: Accelerating Long-Context LLM Inference via Hybrid-Head Sparse Decoding"
_ICLR.cc/2026/Conference — ICLR 2026 Poster_

### Official Review · Reviewer_oLtn · 2025-10-17

**Soundness:** 3
**Presentation:** 4
**Contribution:** 4
**Rating:** 6
**Confidence:** 4

**Summary:**

This paper proposes LycheeDecode, a head-wise sparse attention mechanism with head sampling using HardKuma probability. During training, the HardKuma takes the input of $\alpha, \beta$, which are trainable parameters, to generate the probability focused at 0 and 1 for head selection between full attention and sparse attention. The pattern of sparse attention is propagated across layers from the full attention in shallow layers to the sparse attention in deep layers. The training loss of LycheeDecode composes of two parts: the distillation loss (between the sparsified model and the original dense model) and the regularization loss (to limit the number of full attention heads). LycheeDecode is benchmarked on various datasets, including general-purpose long-context evaluation such as LongBench and RULER, and complex reasoning datasets such as AIME and Olympiad-Bench. LycheeDecode is able to outperform the baselines on all tested benchmarks.

**Strengths:**

1. The paper is well written. The statement of the problem and the description of the methods, especially the HardKuma part, are clear and easy to follow.

2. The design of LycheeDecode is reasonable, efficient, and effective. In DuoAttention, head selection is conducted between the full attention head and the sliding window head with an attention sink. Therefore, the ability to capture long-context dependencies is reduced, since the sliding window has a relatively weaker ability to memorize context history. However, in LycheeDecode, this problem is well solved by using sparse attention instead of the sliding window. The selection process is performed by the head in the same position in previous layers, so no significant overhead for generating sparse patterns is introduced. This design is reasonable and makes a valuable contribution.

3. Mainstream evaluation benchmark datasets are included in this paper, and LycheeDecode achieves promising results on these benchmarks.

**Weaknesses:**

1. Since DuoAttention [1] is an important preliminary work for this paper, comparisons between LycheeDecode and DuoAttention should be conducted on long-context benchmarks such as RULER and LongBench.

2. "By optimizing the distributional parameters of HardKuma during training, our model learns a near-binary selection mechanism directly, thus mitigating the train-inference discrepancy and leading to a more stable and robust head specialization." (line 211-213) During inference, a head is identified as a retrieval head if $\mathbb{E}[z_h] \geq 0.5$. In this case, since $z_h$ is generated from a probability distribution, there is still a chance that this head will be treated as a non-retrieval head during training. Could the authors provide more explanation on why such a design can eliminate the train–inference discrepancy? If possible, a formal proof would help clarify this point.

---

[1] DuoAttention: Efficient Long-Context LLM Inference with Retrieval and Streaming Heads

**Questions:**

See above.

---

> ### Author Response · Authors · 2025-11-24
> **Response to Reviewer oLtn**
>
> Dear Reviewer oLtn,
>
> We sincerely thank the reviewer for the constructive comments and suggestions, which are very helpful for improving our paper. We are also grateful that you recognised the strengths of our paper. We have updated the manuscript accordingly and kindly invite the reviewer to check the revised version for details. Please kindly find point-to-point responses below.
>
> >**Weakness 1**: Since DuoAttention [1] is an important preliminary work for this paper, comparisons between LycheeDecode and DuoAttention should be conducted on long-context benchmarks such as RULER and LongBench.
>
> Thank you for this valuable suggestion. We have conducted a comprehensive comparison with DuoAttention [1] on the LongBench benchmark using the Llama-3-8B-Instruct-1048k model. The results, now included in the revised paper, are summarized below. LycheeDecode outperforms DuoAttention, even when configured with a smaller retrieval head budget.
>
> | Method (Retrieval Head budget) | Avg Score |
> | :--- | :--- |
> | DuoAttention (12.5%) | 25.41 |
> | DuoAttention (25.0%) | 29.15 |
> | LycheeDecode (12.5%) | **33.07** |
>
> We attribute this performance advantage to a key architectural distinction. DuoAttention employs an eviction-based strategy, compressing the KV cache by discarding tokens to reduce memory usage, which can risk losing long-range information. In contrast, LycheeDecode utilizes a selection-based approach: our "Sparse Heads" operate on the full KV cache but compute attention only on a propagated subset of critical tokens. This design preserves the complete global context, leading to superior performance on context-intensive tasks.
>
> To illustrate the different design trade-offs, we also benchmarked latency and memory:
>
> | Metric | Context Length | 16k | 32k | 64k | 128k |
> | :--- | :--- | :--- | :--- | :--- | :--- |
> | **TPOT (ms)** | DuoAttention | 34.91 | 35.47 | 36.03 | 36.76 |
> | | LycheeDecode | 26.43 | 26.68 | 28.41| 29.72 |
> | **Peak Memory (GB)** | DuoAttention | 17.46 | 18.23 | 19.75 | 22.81 |
> | | LycheeDecode | 17.58 | 19.34 | 23.43 | 31.42 |
>
> The results highlight a clear trade-off. LycheeDecode consistently achieves lower latency, while DuoAttention's eviction strategy expectedly yields greater memory efficiency. This comparison clarifies that LycheeDecode is optimized for higher accuracy and speed by preserving context, whereas DuoAttention prioritizes a minimal memory footprint.
>
>
> >**Weakness 2**: "By optimizing the distributional parameters of HardKuma during training, our model learns a near-binary selection mechanism directly, thus mitigating the train-inference discrepancy and leading to a more stable and robust head specialization." (line 211-213) During inference, a head is identified as a retrieval head if $E[z_h] \ge 0.5$. In this case, since is generated from a probability distribution, there is still a chance that this head will be treated as a non-retrieval head during training. Could the authors provide more explanation on why such a design can eliminate the train–inference discrepancy? If possible, a formal proof would help clarify this point.
>
> This is an excellent question that gets to the core of our technical contribution. The key distinction of our approach is that we do not optimize a continuous gating variable directly; rather, we optimize the distributional parameters ($\alpha, \beta$) that drive a near-binary random variable. In methods that learn a continuous scalar, a value like $0.6$ might be optimal for minimizing training loss, but rounding it to $1$ during inference introduces a significant numerical jump.
>
> Our framework mitigates this by shaping the probability distribution itself. Through the combination of the distillation loss and the sparsity regularization term (Equation 7 in the paper), the model is incentivized to adjust $\alpha$ and $\beta$ such that the probability mass of the HardKuma distribution becomes highly concentrated at the boundaries 0 and 1. As visualized in Figure 7 (Appendix D), as training progresses, the probability density function (PDF) for each head collapses towards either 0 or 1.
>
> This concentration ensures that the expected value $E[z_h]$ naturally converges to essentially 0 or 1. Consequently, when we apply the deterministic threshold $E[z_h] \ge 0.5$ during inference, the resulting binary decision aligns closely with the stochastic samples drawn during training (which were already landing on 0 or 1 with high probability). This minimizes the statistical distance between the training behavior and the inference behavior, providing a smoother optimization path compared to rounding a raw continuous variable.
>
> ---
> [1] Xiao G et al. DuoAttention: Efficient Long-Context LLM Inference with Retrieval and Streaming Heads. ICLR 2025.

---

> > ### Comment · Reviewer_oLtn · 2025-11-24
> >
> > Thanks for providing supplementary empirical results. I decide to maintain my score.
> >
> > Another suggestion on explaining Weakness 2 more clearly: I think a better way to present this is to compare the heat map in Figure 7 with other methods, such as DuoAttention or other methods in Table 3. I'm expecting to see that other methods are "greyer" than the proposed method, if my understanding is correct.

---

> > > ### Author Response · Authors · 2025-11-26
> > > **Response to Reviewer oLtn**
> > >
> > > We truly appreciate your constructive suggestion.
> > >
> > > We have updated **Appendix D** and **Figure 7** in the revised manuscript to include a direct comparison with DuoAttention using the Llama-3-8B-Instruct-1048k model.
> > >
> > > The results confirm that observable "grey" regions exist in the DuoAttention heatmap, even at the final training step. This indicates that a subset of attention heads failed to effectively specialize (binarize). In contrast, our method demonstrates a decisive convergence to 0 or 1. This visual comparison effectively validates that the HardKuma distribution minimizes the train-inference discrepancy by enforcing clear head specialization.
> > >
> > > Thank you again for helping us improve the clarity and quality of our work.

---

> > > > ### Comment · Reviewer_oLtn · 2025-11-27
> > > >
> > > > Thank you for providing the comparison in Appendix D and Figure 7. The "grey" regions are very persuasive and show the benefit of using HardKuma distribution.

---

### Official Review · Reviewer_GqdT · 2025-10-26

**Soundness:** 2
**Presentation:** 3
**Contribution:** 2
**Rating:** 4
**Confidence:** 4

**Summary:**

This paper introduces LycheeDecode, a novel method for accelerating long-context inference by leveraging a hybrid sparse attention mechanism. The core idea is to specialize attention heads into retrieval heads and sparse heads". To enable end-to-end training and robust head specialization, the authors propose the HardKuma distribution, a reparameterizable method for near-binary head selection that mitigates the train-inference discrepancy common in discrete optimization. Results on several datasets demonstrates effectiveness.

**Strengths:**

1. The paper is easy to follow.
2. The authors give thorough experiments to validate the proposed methods.

**Weaknesses:**

1. The proposed method does not appear novel to me. For example, the idea of parameterizing discrete variables has been explored in prior work. Techniques such as Gumbel softmax and STE have been widely used.
2. Regarding the motivation in Figure 1, I question the significance of showing the overlap rate between corresponding heads in adjacent layers, as attention heads have no inherent ordering, and heads have no direct correspondence across layers.
3. I am uncertain about the necessity of the HardKuma distribution to reduce the train-inference gap. As I understand it, its parameters are optimized during training, but head types are determined at inference using a fixed threshold—this still introduces a train-inference discrepancy.
4. The method assigns fixed head types regardless of input, which may be suboptimal. A more robust approach would make head specialization adaptive to the specific input during inference.

**Questions:**

Please see the weaknesses.

---

> ### Author Response · Authors · 2025-11-24
> **(1/2) Response to Reviewer GqdT**
>
> Dear Reviewer GqdT,
>
> Thank you for your thoughtful and constructive feedback and comments! We deeply appreciate your suggestions and spare no effort during this response stage to make improvements accordingly. We hope our responses below could address your concerns:
>
> >**Weakness 1**: The proposed method does not appear novel to me. For example, the idea of parameterizing discrete variables has been explored in prior work. Techniques such as Gumbel softmax and STE have been widely used.
>
> We acknowledge that parameterizing discrete choices via differentiable relaxation is an established technique. Our contribution lies in the novel application and effective integration of the **Hard Kumaraswamy (HardKuma) distribution** within a hybrid-head framework specifically for long-context inference. We find that HardKuma is particularly well-suited for this task due to its properties: it generates samples on the closed interval $[0, 1]$ with probability mass that naturally concentrates at the endpoints. This provides a more stable and direct optimization path for learning the near-binary decision of classifying head types, which we demonstrate leads to strong empirical results.
>
> >**Weakness 2**: Regarding the motivation in Figure 1, I question the significance of showing the overlap rate between corresponding heads in adjacent layers, as attention heads have no inherent ordering, and heads have no direct correspondence across layers.
>
> We agree that attention heads do not possess a fixed, pre-existing functional identity based on their index. We would like to clarify that the primary purpose of Figure 1 was not to assert such a relationship. Instead, the visualization was intended as an analytical tool to illustrate the key phenomenon that motivates our work.
>
> The index-based comparison presented in the heatmap is designed to be consistent with the information propagation mechanism within our proposed LycheeDecode framework. Specifically, during inference, a "sparse head" at layer $l$ reuses the set of critical tokens identified by the head at the very same index in the preceding layer $l-1$ (if that head was a "retrieval head"). Therefore, Figure 1 is designed to demonstrate that under this specific, index-aligned sharing paradigm, the degree of token overlap varies significantly across different head indices. This high variance underscores our central argument: a uniform, layer-wise sharing strategy is suboptimal as it overlooks this functional diversity, thereby justifying the need for our more fine-grained, head-level specialization strategy.
>
> To further validate that our method does not depend on a pre-existing head correspondence, we conducted an ablation study where we randomly permuted the attention head weights within each layer of the pre-trained model before applying our specialization training. The resulting models achieved nearly identical performance on LongBench (differing by less than 3%), indicating that our method does not rely on a specific head correspondence. Consequently, we adopted the more straightforward implementation without permutation, as it offers greater simplicity and reproducibility without compromising performance.
>
> Finally, it is worth noting that this index-based, layer-to-layer information transfer is also a practical convention in related work. For instance, hierarchical sharing methods like TidalDecode [1] also propagate the set of key tokens identified by each head to the head at the same index in the subsequent layer for reuse. Therefore, our implementation is consistent with existing practices in the field.
>
> >**Weakness 3**: I am uncertain about the necessity of the HardKuma distribution to reduce the train-inference gap. As I understand it, its parameters are optimized during training, but head types are determined at inference using a fixed threshold—this still introduces a train-inference discrepancy.
>
> In our framework, we do not simply optimize a continuous gating value; rather, we optimize the shape parameters $(\alpha, \beta)$ of the HardKuma distribution. Driven by the sparsity regularization term in our loss function, the optimization process forces the probability density function (PDF) to collapse towards the boundaries of 0 and 1. As visualized in Appendix D (Figure 7), the distribution for each head becomes highly polarized during training.
>
> As a result, the expected value $E[z]$ naturally converges to a value very close to either 0 or 1. When we apply a deterministic threshold (e.g., $E[z] > 0.5$) at inference, this decision closely aligns with the stochastic samples drawn during training (which were already landing near 0 or 1 with high probability). In contrast, methods that learn a raw continuous variable may result in values clustered mid-range (e.g., 0.4 or 0.6), where rounding introduces a larger discrepancy. HardKuma inherently shapes the distribution to be near-binary, thus reducing this gap.

---

> ### Author Response · Authors · 2025-11-24
> **(2/2) Response to Reviewer GqdT**
>
> >**Weakness 4**: The method assigns fixed head types regardless of input, which may be suboptimal. A more robust approach would make head specialization adaptive to the specific input during inference.
>
> We agree that an ideal system would dynamically adapt head roles based on the specific input. However, implementing a fully input-adaptive mechanism would require a dynamic router or predictor to determine head types at runtime for every token. This introduces additional computational overhead and memory access costs, which could negate the speed gains achieved by sparse attention.
>
> Our approach uses a static assignment to strike a practical balance between performance and inference speed. We hypothesize that attention heads in pre-trained LLMs already possess implicit, specialized functional tendencies. Our method aims to identify and formalize these intrinsic roles. By keeping head types fixed post-training, LycheeDecode enables a zero-overhead lookup during inference, maximizing acceleration.
>
> ---
> [1] Yang L et al. TidalDecode: Fast and Accurate LLM Decoding with Position Persistent Sparse Attention. ICLR 2025.

---

### Official Review · Reviewer_Zofo · 2025-10-31

**Soundness:** 3
**Presentation:** 3
**Contribution:** 3
**Rating:** 6
**Confidence:** 4

**Summary:**

The paper proposes LycheeDecode, a sparse attention mechanism to accelerate long-context LLM inference by partitioning heads into a small set of "Retrieval Heads" that perform full attention and a majority of "Sparse Heads" that efficiently reuse the critical tokens identified by the retrieval heads. To learn this specialization, it introduces the HardKuma distribution, which mitigates the train-inference discrepancy common in discrete optimization. Experiments show LycheeDecode achieves up to 2.7x speedup at 128K context lengths while matching or even exceeding the performance of the full-attention baseline.

**Strengths:**

1. The paper's primary innovation is the hybrid-head decoding mechanism. This mechanism establishes a head-indexed pipeline, where a Retrieval Head in one layer selects tokens specifically for its corresponding Sparse Head in the next layer to reuse.

2. The use of the HardKuma distribution directly targets a known weakness (train-inference discrepancy) in prior training-based specialization methods, leading to a more stable and direct optimization of the discrete head roles.

**Weaknesses:**

1. Missing Key Baseline Comparisons: The paper's central claim is that its cooperative head-specialization architecture is superior. However, it fails to provide any end-to-end performance or speed comparisons against the most direct SOTA competitors in the head-specialization sub-field (e.g., DuoAttention, RazorAttention). It only compares against a layer-sharing method (TidalDecode). This makes the SOTA claim unsubstantiated, as we cannot see how it performs against other architectures with the same design philosophy.

2. Weak Evidence: The claim of surpassing the full-attention baseline is remarkable. The paper's only explanation is a vague hypothesis about "filtering out irrelevant context that may act as noise." This claim requires rigorous qualitative proof (e.g., visualizations, case studies), which is absent.

3. Lack of Sensitivity Analysis: The method's performance hinges on two key hyperparameters: the retrieval head budget ($N_{target}$) and the token budget ($k$). The paper provides no ablation studies on how varying these budgets affects the performance/latency trade-off. The $N_{target}$ value was just set to "match... TidalDecode," which is arbitrary and likely not optimal.

**Questions:**

1. Missing Baselines (Duo/Razor): Why are there no end to end comparisons against DuoAttention and RazorAttention? The paper's premise is that its cooperative architecture is a key advantage over DuoAttention's isolated head roles, but this central claim is not experimentally validated.

2. Confounding Variable (Cache Correction): In Table 2, the "Cache Correction" (CC) strategy provides a large performance boost. Did you test a "Full Attention + CC" baseline? Your current results could be interpreted as the CC method being the main source of the performance gain over the baseline, not LycheeDecode.

3. Evidence for "Surpassing Full Attention": Can you provide a specific, qualitative example (e.g., visualizing attention) that proves the full-attention model failed due to "noise" while LycheeDecode succeeded by filtering it?

---

> ### Author Response · Authors · 2025-11-24
> **(1/2) Response to Reviewer Zofo**
>
> Dear Reviewer Zofo,
>
> Thank you for your thoughtful and constructive feedback and comments! We deeply appreciate your suggestions and spare no effort during this response stage to make improvements accordingly. We hope our responses below could address your concerns:
>
> >**Question 1**: Missing Baselines (Duo/Razor): Why are there no end to end comparisons against DuoAttention and RazorAttention? The paper's premise is that its cooperative architecture is a key advantage over DuoAttention's isolated head roles, but this central claim is not experimentally validated.
>
> >**Weakness 1**: Missing Key Baseline Comparisons: The paper's central claim is that its cooperative head-specialization architecture is superior. However, it fails to provide any end-to-end performance or speed comparisons against the most direct SOTA competitors in the head-specialization sub-field (e.g., DuoAttention, RazorAttention). It only compares against a layer-sharing method (TidalDecode). This makes the SOTA claim unsubstantiated, as we cannot see how it performs against other architectures with the same design philosophy.
>
>
> Thank you for emphasizing the need for this comparison. We have conducted a detailed end-to-end comparison with DuoAttention on the Llama-3-8B-Instruct-1048k model and integrated the results into the revised paper to substantiate our SOTA claims.
>
> In terms of generative quality (LongBench Average Score), LycheeDecode clearly outperforms DuoAttention. Our method with just **12.5%** retrieval heads achieves a higher score than DuoAttention configured with a **25.0%** budget. This performance advantage stems from our novel **selection-based** design. DuoAttention reduces memory by permanently evicting tokens, which risks information loss. In contrast, LycheeDecode maintains the full KV cache and uses "Sparse Heads" to efficiently compute attention on a propagated subset of critical tokens, preserving the global context.
>
> | Method (Retrieval Head budget) | Avg Score |
> | :--- | :--- |
> | DuoAttention (12.5%) | 25.41 |
> | DuoAttention (25.0%) | 29.15 |
> | **LycheeDecode (12.5%)** | **33.07** |
>
> In terms of decoding speed, LycheeDecode also demonstrates a distinct advantage, achieving faster generation (lower TPOT) across all evaluated context lengths, thanks to our custom hybrid-head kernels.
>
> | Metric | Context Length | 16k | 32k | 64k | 128k |
> | :--- | :--- | :--- | :--- | :--- | :--- |
> | **TPOT (ms)** | DuoAttention | 34.91 | 35.47 | 36.03 | 36.76 |
> | | LycheeDecode | 26.43 | 26.68 | 28.41| 29.72 |
> | **Peak Memory (GB)** | DuoAttention | 17.46 | 18.23 | 19.75 | 22.81 |
> | | LycheeDecode | 17.58 | 19.34 | 23.43 | 31.42 |
>
> While DuoAttention offers better memory efficiency due to its KV cache compression, our results confirm that LycheeDecode establishes a new SOTA trade-off favoring accuracy and speed.
>
> >**Question 2**: Confounding Variable (Cache Correction): In Table 2, the "Cache Correction" (CC) strategy provides a large performance boost. Did you test a "Full Attention + CC" baseline? Your current results could be interpreted as the CC method being the main source of the performance gain over the baseline, not LycheeDecode.
>
> That is a fair point. The Cache Correction (CC) strategy is designed specifically to mitigate error accumulation in *sparse* attention methods by periodically refreshing the KV cache. While we understand the rationale behind the suggestion, we believe a "Full Attention + CC" baseline may not offer additional insights because the Full Attention model performs exact computation at every step and does not suffer from the sparse approximation errors that CC is designed to rectify.
>
> More importantly, our results in Table 2 show that CC is not the sole driver of performance. Even **without** Cache Correction, on the DeepSeek-R1-Distill-Llama-8B model, LycheeDecode achieves an average score of **36.8**, which is higher than both the Full Attention baseline (35.4) and TidalDecode (31.6). This demonstrates the inherent strength of our hybrid-head architecture.

---

> ### Author Response · Authors · 2025-11-24
> **(2/2) Response to Reviewer Zofo**
>
> >**Question 3**: Evidence for "Surpassing Full Attention": Can you provide a specific, qualitative example (e.g., visualizing attention) that proves the full-attention model failed due to "noise" while LycheeDecode succeeded by filtering it?
>
> >**Weakness 2**: Weak Evidence: The claim of surpassing the full-attention baseline is remarkable. The paper's only explanation is a vague hypothesis about "filtering out irrelevant context that may act as noise." This claim requires rigorous qualitative proof (e.g., visualizations, case studies), which is absent.
>
> We thank you for pushing for more rigorous evidence. We agree that this claim requires more than a high-level hypothesis. To substantiate our "noise filtering" explanation, we have conducted a new qualitative analysis and added it to **Appendix E.5** of the revised manuscript.
>
> In this analysis, we provide a case study with a reasoning prompt that includes irrelevant distractor text. We visualize the attention distribution of the final answer token, comparing the average attention from Retrieval Heads (which perform full attention) against that from Sparse Heads. The visualizations clearly show that while Retrieval Heads assign weight to the noisy tokens, the Sparse Heads, which constitute the majority of the model's computation, effectively ignore this noise by design, as these tokens were not propagated as critical. This "denoising" effect allows the model to focus on the relevant reasoning path, providing a concrete example of how our method can outperform the full-attention baseline.
>
> >**Weakness 3**: Lack of Sensitivity Analysis: The method's performance hinges on two key hyperparameters: the retrieval head budget ($N_{target}$) and the token budget ($k$). The paper provides no ablation studies on how varying these budgets affects the performance/latency trade-off. The value was just set to "match... TidalDecode," which is arbitrary and likely not optimal.
>
> We thank the reviewer for this valuable feedback. We have conducted a new experiment and included a detailed analysis in the revised manuscript (Appendix E.4). In this study, we explore a range of sparsity configurations by varying two key hyperparameters: the critical token budget ($k\in\{1024, 2048, 4096\}$) and the ratio of retrieval heads (12.5%, 25.0%, and 50.0% of the total attention heads).
>
> We evaluated the performance based on the average score across the LongBench benchmark. Efficiency was quantified by the end-to-end decoding speedup (measured via Time Per Output Token, TPOT) relative to the Full Attention baseline. The results are summarized in the tables below.
>
> **Table A: Performance (Average Score on LongBench)**
>
> |  | 12.5% | 25.0% | 50.0% |
> | :--- | :---: | :---: | :---: |
> | **1024** | 30.59 | 31.06 | 32.68 |
> | **2048** | 31.90 | 32.96 | 32.84 |
> | **4096** | 32.89 | 33.92 | 33.53 |
>
> **Table B: Efficiency (End-to-End Decoding Speedup vs. Full Attention)**
>
> |  | 12.5% | 25.0% | 50.0% |
> | :--- | :---: | :---: | :---: |
> | **1024** | 1.31 | 1.17 | 1.05 |
> | **2048** | 1.27 | 1.12 | 1.02 |
> | **4096** | 1.24 | 1.10 | 0.99 |
>
> These results clearly illustrate a Pareto frontier between performance and efficiency. As expected, a larger token budget or a higher proportion of retrieval heads improves performance by enabling the model to capture more complex, long-range dependencies through greater contextual access and more precise attention computations. Conversely, these gains correspond to a reduction in decoding speed due to the increased computational load.
>
> Notably, with token budgets of 2048 and 4096, the configuration with 25% retrieval heads actually yields superior average scores compared to using 50% retrieval heads. We hypothesize that this result stems from the noise-filtering capability of sparse heads. By restricting more heads to attend only to critical tokens, the model effectively filters out irrelevant context that might otherwise act as noise, thereby enhancing performance.
>
> ---
> [1] Xiao G et al. DuoAttention: Efficient Long-Context LLM Inference with Retrieval and Streaming Heads. ICLR 2025.

---

### Official Review · Reviewer_ZVhw · 2025-10-31

**Soundness:** 3
**Presentation:** 2
**Contribution:** 3
**Rating:** 6
**Confidence:** 4

**Summary:**

This paper introduces LycheeDecode, which learns to partition attention heads into a few retrieval heads (full attention with top-k token selection) and many sparse heads (reusing selected tokens). The method uses HardKuma to mitigate train–inference mismatch and employs a custom TileLang block-sparse kernel. On long-context benchmarks (LongBench, RULER) and reasoning tasks (AIME24, OlympiadBench), it achieves competitive accuracy with up to 2.7× speedup at 128k context length.

**Strengths:**

1. The paper provides compelling motivation that layer-level token sharing ignores significant functional diversity among heads, with empirical evidence showing highly variable top-k overlap across adjacent layers.
2. Trainable retrieval and sparse head assignment, HardKuma offers a principled, differentiable relaxation that tends toward binary outcomes without rounding.
3. The custom hybrid-head kernel yields substantial kernel-level improvements, this strengthens the practicality claim.

**Weaknesses:**

1. Lack of comparison with training-based long-context inference, DuoAttention is arguably the most directly comparable prior work, and the omission makes it difficult to isolate novelty beyond the use of top-k propagation.
2. Head role consistency and interpretability are not evaluated, how stable head assignments are across different settings (random setting), whether specialization generalizes to unseen domains.
3. Training complexity is under-characterized. The method computes both full and sparse attention per head during training, doubling attention paths. FLOPs, KV bandwidth, and convergence cost are not reported. Scaling to deeper models remains unclear.
4. Propagation across layers may accumulate error.

**Questions:**

1. Do retrieval heads discovered by LycheeDecode overlap with previous methods' identified retrieval heads? (such as DuoAttention or QR head https://arxiv.org/abs/2506.09944)
2. What is the training FLOPs overhead vs. DuoAttention or RazorAttention?
3. Will head roles transfer to long-form summarization or narrative QA?
4. How does training complexity grow with more layers and longer sequences?
5. What is the Jaccard overlap between retrieval heads discovered by LycheeDecode and DuoAttention? Is the retrieval-head model-dependent or training method related?

---

> ### Author Response · Authors · 2025-11-24
> **(1/2) Response to Reviewer ZVhw**
>
> Dear Reviewer ZVhw,
>
> We sincerely thank you for the constructive comments and suggestions, which are very helpful for improving our paper. We are also grateful that you recognized the strengths of our paper. Please kindly find point-to-point responses below.
>
> >**Weakness 1**: Lack of comparison with training-based long-context inference, DuoAttention is arguably the most directly comparable prior work, and the omission makes it difficult to isolate novelty beyond the use of top-k propagation.
>
> We thank you for identifying DuoAttention [1] as a key point of comparison. better highlight the novelty of LycheeDecode, we have added a comprehensive comparison on the LongBench benchmark. The core novelty of our work lies in the architectural shift from an **eviction-based** to a **selection-based** paradigm.
>
> *   **DuoAttention (Eviction-based):** Its "Streaming Heads" compress the KV cache by permanently discarding tokens. This design choice effectively reduces the memory footprint but may limit access to long-range dependencies.
> *   **LycheeDecode (Selection-based):** Our approach maintains the full KV cache, ensuring complete contextual information is always available. "Sparse Heads" cooperatively reuse a curated subset of critical tokens identified by "Retrieval Heads," enabling both efficiency and high-precision attention.
>
> This architectural distinction leads to clear quantitative advantages. LycheeDecode demonstrates stronger performance on LongBench, even with a smaller budget of retrieval heads compared to DuoAttention.
>
> | Method (Retrieval Head budget) | Avg Score |
> | :--- | :--- |
> | DuoAttention (12.5%) | 25.41 |
> | DuoAttention (25.0%) | 29.15 |
> | **LycheeDecode (12.5%)** | **33.07** |
>
> Furthermore, our evaluation of inference metrics highlights the different trade-offs each method makes:
>
> | Metric | Context Length | 16k | 32k | 64k | 128k |
> | :--- | :--- | :--- | :--- | :--- | :--- |
> | **TPOT (ms)** | DuoAttention | 34.91 | 35.47 | 36.03 | 36.76 |
> | | **LycheeDecode** | **26.43** | **26.68** | **28.41**| **29.72** |
> | **Peak Memory (GB)** | DuoAttention | 17.46 | 18.23 | 19.75 | 22.81 |
> | | LycheeDecode | 17.58 | 19.34 | 23.43 | 31.42 |
>
> While DuoAttention is optimized for a minimal memory footprint, LycheeDecode's cooperative selection mechanism is optimized for lower latency and superior accuracy. This validates that our approach offers a novel and effective alternative to eviction-based methods.
>
> >**Question 1**: Do retrieval heads discovered by LycheeDecode overlap with previous methods' identified retrieval heads? (such as DuoAttention or QR head https://arxiv.org/abs/2506.09944)
>
> >**Question 5**: What is the Jaccard overlap between retrieval heads discovered by LycheeDecode and DuoAttention? Is the retrieval-head model-dependent or training method related?
>
> This is an insightful question. We analyzed the overlap between the retrieval heads identified by LycheeDecode and DuoAttention for the Llama-3-8B-Instruct-1048k model. We found that **62.5%** of the heads identified as "retrieval" by LycheeDecode are also classified as "retrieval" by DuoAttention, corresponding to a **Jaccard overlap of 45.5%**.
>
> The overlap rate is not particularly high because: (1) The training methods differ (we optimize HardKuma distribution parameters, whereas DuoAttention optimizes a continuous gating variable); (2) The classification taxonomy differs (Retrieval vs. Sparse in ours; Retrieval vs. Streaming in DuoAttention).
>
> Additionally, while retrieval heads vary across different models, the same model using different training methods has a high overlap rate of retrieval heads. For instance, in Llama-3-8B-Instruct-1048k, the overlap between retrieval heads identified by HardKuma and those found by directly optimizing gating variables exceeds 70%.
>
>
> >**Question 2**: What is the training FLOPs overhead vs. DuoAttention or RazorAttention?
>
> Compared to DuoAttention’s approach of directly optimizing gating variables, our additional overhead primarily stems from the HardKuma sampling process (inverse CDF transformation, stretching, and rectification). These steps involve only a small number of **scalar operations**, making their computational complexity independent of sequence length. Compared to the substantial overhead of the attention mechanism itself, the cost of this sampling process is low.
>
> >**Question 4**: How does training complexity grow with more layers and longer sequences?
>
> The training complexity of our method grows linearly with the number of layers and quadratically with the sequence length, which is consistent with the standard Transformer architecture.

---

> ### Author Response · Authors · 2025-11-24
> **(2/2) Response to Reviewer ZVhw**
>
> >**Weakness 3**: Training complexity is under-characterized. The method computes both full and sparse attention per head during training, doubling attention paths. FLOPs, KV bandwidth, and convergence cost are not reported. Scaling to deeper models remains unclear.
>
> During training, we compute both full and sparse attention paths (Eq. 5). However, since the **LLM backbone parameters are frozen**, we only optimize the scalar parameters $(\alpha, \beta)$ for each head's HardKuma distribution. This lightweight process converges rapidly: training completes in just 3000 steps on a single NVIDIA A100 GPU, taking **less than 4 hours**. The computational overhead from the dual attention paths is approximately 1.3× that of standard attention, as the sparse path operates on a token subset (30% of sequence length in our training). Similarly, while the full attention path accesses the entire KV cache, the sparse attention path retrieves K/V states only for the selected tokens, resulting in a KV cache bandwidth usage that is also 1.3× that of standard attention.
>
> >**Question 3**: Will head roles transfer to long-form summarization or narrative QA?
>
> Yes, our experiments show strong generalization. The head specialization was learned using a passkey retrieval task on the BookSum dataset, yet the specialized model performs well on a wide range of unseen downstream tasks. As detailed in Table 1, LycheeDecode achieves competitive performance on the LongBench benchmark, which includes diverse tasks like **NrtQA** (NarrativeQA) and **QMSum** (Summarization). This successful transfer indicates that the identified "Retrieval Heads" capture fundamental model behaviors rather than overfitting to the training objective.
>
> >**Weakness 2**: Head role consistency and interpretability are not evaluated, how stable head assignments are across different settings (random setting), whether specialization generalizes to unseen domains.
>
> We evaluated stability by training with different random seeds and various initialization strategies for the HardKuma parameters $\alpha$ and $\beta$ (all-ones, all-zeros, and random initialization). We found that the identified retrieval heads remained consistent across all these settings, demonstrating the robustness of our head specialization process.The generalization capability, as discussed in response for Question 3, further supports that our method identifies meaningful and transferable functional roles.
>
> >**Weakness 4**: Propagation across layers may accumulate error.
>
> Our architecture is designed to mitigate error accumulation. The set of critical tokens is not propagated indefinitely. Instead, "Retrieval Heads" are distributed throughout the model and periodically **refresh** this set by performing full attention over the entire context. This re-evaluation prevents the compounding of approximation errors from sparse attention. Empirically, our strong performance, particularly in complex reasoning tasks (Table 2) where LycheeDecode's performance is competitive with the full-attention baseline on the DeepSeek-R1-Distill-Llama-8B model, suggests that error accumulation is effectively managed.
>
> ---
> [1] Xiao G et al. DuoAttention: Efficient Long-Context LLM Inference with Retrieval and Streaming Heads. ICLR 2025.

---

### Official Review · Reviewer_vCSt · 2025-11-02

**Soundness:** 3
**Presentation:** 3
**Contribution:** 2
**Rating:** 4
**Confidence:** 4

**Summary:**

This paper proposes LycheeDecode, a sparse attention method that categorizes attention heads into "retrieval heads" and "sparse heads" to accelerate long-context LLM inference. The retrieval heads perform full attention to identify critical tokens, which are then reused by sparse heads for efficient computation.

**Strengths:**

The paper presents a well-motivated approach. The experimental results demonstrate that LycheeDecode achieves performance comparable to full attention baselines on complex reasoning tasks.

**Weaknesses:**

1. Recent work has demonstrated that trainable sparse attention can also achieve efficient decoding [1-3]. The paper lacks discussion and empirical comparison with these methods.

[1] Native sparse attention: Hardware-aligned and natively trainable sparse attention
[2] Minicpm4: Ultra-efficient llms on end devices
[3] SeerAttention-R: Sparse Attention Adaptation for Long Reasoning

2. While the paper shows kernel-level speedup for different sparse head ratios, there is no corresponding analysis of how varying the proportion of retrieval heads affects model performance. An ablation study examining the performance-efficiency trade-off across different retrieval head budgets would strengthen the paper.

3. The efficiency evaluation focuses on synthetic settings with fixed context lengths. For math reasoning tasks that involve long chain-of-thought generation, where the sequence length grows dynamically during decoding, end-to-end latency measurements would be more convincing.

**Questions:**

Please refer to Weaknesses.

---

> ### Author Response · Authors · 2025-11-24
> **(1/2) Response to Reviewer vCSt**
>
> Dear Reviewer vCSt,
>
> Thanks for your comprehensive and detailed suggestions for our work! We really value your comment on the applicability and practical application of our conclusions. We hope our response could address your concerns:
>
> >**Weakness 1**: Recent work has demonstrated that trainable sparse attention can also achieve efficient decoding [1-3]. The paper lacks discussion and empirical comparison with these methods.
>
> Thank you for highlighting these relevant works. We would like to clarify the distinctions between our approach and methods like Native Sparse Attention [1] (NSA) and MiniCPM4 [2]. Both NSA and MiniCPM4 represent a paradigm that requires extensive post-training to adapt the model to sparse structures. In contrast, LycheeDecode focuses on the **lightweight identification** of retrieval heads in existing pre-trained models, which involves lower computational costs. Furthermore, NSA and MiniCPM4 are often tightly coupled with specific architectures (e.g., MiniCPM4 uses custom kernels for Group-Size-16 GQA, and NSA is bound to DeepSeek models), making fair and direct comparisons on general models like Llama3 or Qwen3 experimentally challenging.
>
> Regarding SeerAttention-R [3], which uses a trainable gating network, we have conducted a comparison on LongBench using the Qwen3-8B model in the revised manuscript (Table 1).
>
> | Budget | SeerAttention-R | LycheeDecode |
> | :--- | :---: | :---: |
> | 1024 | 31.71 | 32.38 |
> | 4096 | 33.38 | 33.48 |
>
>  While SeerAttention-R achieves performance comparable to LycheeDecode, it relies on a trainable gating network to predict important tokens. Training such a network generally entails higher complexity compared to our lightweight head identification process, which only optimizes scalar distribution parameters. We incorporate the discussion and the qualitative comparison of SeerAttention-R into the main text and add the papers you mentioned to reference.
>
> >**Weakness 2**: While the paper shows kernel-level speedup for different sparse head ratios, there is no corresponding analysis of how varying the proportion of retrieval heads affects model performance. An ablation study examining the performance-efficiency trade-off across different retrieval head budgets would strengthen the paper.
>
> We thank the reviewer for this valuable feedback. We have conducted a comprehensive ablation study to address this, which is now included in **Appendix E.4** of the revised manuscript. We investigated the impact of varying the **retrieval head ratio** (12.5%, 25.0%, and 50.0%) across different token budgets (1024, 2048, and 4096).
>
> We evaluated performance by the average score on the LongBench benchmark and efficiency by the end-to-end decoding speedup (measured via Time Per Output Token, TPOT) relative to the Full Attention baseline. The results are summarized below and presented as a Pareto frontier in Figure 8.
>
> **Table A: Performance (Average Score on LongBench)**
>
> |  | 12.5% | 25.0% | 50.0% |
> | :--- | :---: | :---: | :---: |
> | **1024** | 30.59 | 31.06 | 32.68 |
> | **2048** | 31.90 | 32.96 | 32.84 |
> | **4096** | 32.89 | 33.92 | 33.53 |
>
> **Table B: Efficiency (End-to-End Decoding Speedup vs. Full Attention)**
>
> |  | 12.5% | 25.0% | 50.0% |
> | :--- | :---: | :---: | :---: |
> | **1024** | 1.31 | 1.17 | 1.05 |
> | **2048** | 1.27 | 1.12 | 1.02 |
> | **4096** | 1.24 | 1.10 | 0.99 |
>
> These results illustrate a clear trade-off. A smaller ratio of retrieval heads yields the highest decoding speedup by minimizing computational overhead. Regarding performance, we observed that simply maximizing the number of retrieval heads does not always lead to the best results. Notably, with larger token budgets (2048 and 4096), the 25.0% retrieval head configuration outperforms the 50.0% setting. We hypothesize that this is due to the noise-filtering property of our approach; an excessive proportion of retrieval heads may introduce irrelevant context, whereas a balanced configuration allows sparse heads to effectively focus on the most critical information.

---

> ### Author Response · Authors · 2025-11-24
> **(2/2) Response to Reviewer vCSt**
>
> >**Weakness 3**: The efficiency evaluation focuses on synthetic settings with fixed context lengths. For math reasoning tasks that involve long chain-of-thought generation, where the sequence length grows dynamically during decoding, end-to-end latency measurements would be more convincing.
>
> We appreciate this point and agree that reasoning tasks have a dynamic nature. We adopt **TPOT (Time Per Output Token)** as our primary metric (Figure 3), which is a standard for evaluating autoregressive generation efficiency. We use inputs of fixed lengths as a controlled and representative proxy because the TPOT for decoding the $n$-th token in a sequence is equivalent to the TPOT measured with a static input context of length $k+n$, where $k$ is the initial prompt length. This methodology allows us to systematically and reproducibly evaluate the latency at different stages of the decoding process, which would be challenging in a live generation setting due to the variable and uncontrollable length of the generated output.
>
> ---
> [1] NYuan J et al. Native sparse attention: Hardware-aligned and natively trainable sparse attention. ACL 2025.
>
> [2] Team M et al. Minicpm4: Ultra-efficient llms on end devices. arXiv:2506.07900.
>
> [3] Gao Y et al. SeerAttention-R: Sparse Attention Adaptation for Long Reasoning. arXiv:2506.08889.

---

### Author Response · Authors · 2025-11-27
**General Response**

We sincerely thank all the reviewers for their thoughtful comments and constructive suggestions, which significantly helped us strengthen our paper. We are encouraged to see that the reviewers recognize the motivation of our approach (Reviewer vCSt, ZVhw), the novelty of the proposed hybrid-head mechanism and HardKuma distribution (Reviewer ZVhw, Zofo, oLtn), and its comprehensive experimental validation (Reviewer vCSt, GqdT).

In response to the reviewers' feedback, we have submitted an updated version of our paper, which now includes stronger baseline comparisons, deeper ablation studies, and qualitative visualizations. Below, we summarize the revisions made to the paper.

### **Summary of Paper Revision**

*   **[Section 2]** We updated the Related Work section to include recent trainable sparse attention mechanisms (Native Sparse Attention, MiniCPM).
*   **[Section 4.2.1 & Table 1]** We incorporated comparisons with strong baselines, including DuoAttention and SeerAttention-R, on the LongBench benchmark to further validate the effectiveness of LycheeDecode.
*   **[Appendix D & Figure 7]** We updated the training dynamics visualization to include a direct comparison with DuoAttention. This highlights the effectiveness of the HardKuma distribution in achieving decisive head specialization compared to continuous gating variables
*   **[Appendix E.4 & Figure 8]** We added a new ablation study to analyze the trade-offs between model performance and inference efficiency under different retrieval head ratios and token budgets, providing a more detailed sensitivity analysis.
*   **[Appendix E.5 & Figure 9]** We added a qualitative analysis with attention visualizations to demonstrate how Sparse Heads effectively filter out irrelevant context ("noise"), providing empirical evidence for why LycheeDecode can surpass full-attention baselines.

We would like to express our sincere gratitude to all reviewers for their valuable time and effort in helping improve our work, and we would like to check whether you have any additional questions or concerns that we can help clarify.

Best regards,

Authors

---

### Meta-Review · Area_Chair_sTQC · 2026-01-06

**Summary:**

This paper introduces LycheeDecode, a sparse attention mechanism for long-context LLM inference that partitions attention heads into a small set of retrieval heads and many sparse heads to improve efficiency. Reviewers consistently recognize the well-motivated method and the thorough experiment validation with satisfiable performance. Reviewer vCSt raised concerns about missing discussion with related work and experimental analysis, which is turn provided in the author rebuttal. Reviewer GqdT raised concerns about novelty of the proposed method, which (to me) is partially addressed by the rebuttal. Given the acknowledged contributions and strong empirical validation, I lean toward accepting this paper.

**Reviewer Concerns:**

Reviewer vCSt:
- Missing discussion with a recent related work.
- Missing analysis of performance w.r.t. the proportion of retrieval head.
- Request of end-to-end latency measurements for long COT generation.

Reviewer ZVhw:
- Lack of comparison with training-based long-context inference.
- Head role consistency and interpretability.
- Missing discussion of training complexity.

Reviewer Zofo:
- Missing baseline comparison.
- Missing rigorous verifications of surpassing the full-attention baseline.
- Lack of sensitivity analysis.

Reviewer GqdT:
- Novelty: parameterizing discrete variables has been widely explored.
- the necessity of the HardKuma distribution. Reviewer questioned the necessity and author explained that it is for polarized head attention, which makes sense to me. I believe it address the reviewers concern.
- assigns fixed head roles may be suboptimal. Author explain that this is a trade-off between performance and speed, for using an adaptive head roles require more computation.

Reviewer oLtn:
- Missing comparisons with DuoAttention, which is also raised by other reviewers.
- Explaination of elimination of the train–inference discrepancy.

**Reviewer Scores:**

The discussion between reviewers and authors was unexpectedly terminated early. Area Chairs are therefore asked to put ourselves in the reviewers’ shoes and provide our best estimate of how scores might have changed. While this is challenging, the following reflects only the AC’s best guess:

Reviewer vCSt: 4 → 6\
Reviewer ZVhw: 6 → 6\
Reviewer Zofo: 6 → 6\
Reviewer GqdT: 4 → 4\
Reviewer oLtn: 6 → 6

---

### Decision · Program_Chairs · 2026-01-26

Accept (Poster)